# Threat of shock increases excitability and connectivity of the intraparietal sulcus

Nicholas L Balderston[1]*, Elizabeth Hale[1], Abigail Hsiung[1], Salvatore Torrisi[1], Tom Holroyd[2], Frederick W Carver[2], Richard Coppola[2], Monique Ernst[1], Christian Grillon[1]

[1]Section on Neurobiology of Fear and Anxiety, National Institute of Mental Health, National Institutes of Health, Bethesda, United States; [2]MEG Core Facility, National Institute of Mental Health, National Institutes of Health, Bethesda, United States

**Abstract** Anxiety disorders affect approximately 1 in 5 (18%) Americans within a given 1 year period, placing a substantial burden on the national health care system. Therefore, there is a critical need to understand the neural mechanisms mediating anxiety symptoms. We used unbiased, multimodal, data-driven, whole-brain measures of neural activity (magnetoencephalography) and connectivity (fMRI) to identify the regions of the brain that contribute most prominently to sustained anxiety. We report that a single brain region, the intraparietal sulcus (IPS), shows both elevated neural activity and global brain connectivity during threat. The IPS plays a key role in attention orienting and may contribute to the hypervigilance that is a common symptom of pathological anxiety. Hyperactivation of this region during elevated state anxiety may account for the paradoxical facilitation of performance on tasks that require an external focus of attention, and impairment of performance on tasks that require an internal focus of attention.

*For correspondence: nicholas.balderston@nih.gov

**Competing interests:** The authors declare that no competing interests exist.

## Introduction

Current models of anxiety disorders suggest that pathological anxiety results from excessive or inappropriate activation of the same neural circuits that are responsible for adaptive anxiety in the face of threat (*Insel et al., 2010*; *Insel, 2014*). Although there is a long history of translational work studying neural systems mediating the acute fear response (*Pavlov, 1927*; *Fanselow and Poulos, 2005*; *Fullana et al., 2016*), much less is known about the neural systems mediating prolonged periods of elevated state anxiety. Closing this knowledge gap is critical because the occurrence of prolonged periods of elevated state anxiety is one of the primary symptoms of all anxiety disorders (*American Psychiatric Association, 2013*). Therefore, understanding neural mechanisms underlying prolonged periods of elevated anxiety has the potential to identify targets for the treatment of anxiety disorders, which are among the most prevalent psychiatric disorders (*Kessler et al., 2005*).

The gold-standard translational paradigm for studying elevated state anxiety in the laboratory is the threat of shock paradigm, during which subjects are exposed to periods when they are either safe, or at risk for receiving unpredictable aversive electrical stimulations (*Schmitz and Grillon, 2012*; *Grillon, 2008*; *Grillon and Baas, 2003*). It allows for the experimental manipulation of state anxiety within subjects (*Grillon and Baas, 2003*, *1998*; *Grillon et al., 1991*, *2007*, *2008*, *2009*), which can be quantified using psychological and physiological measures (*Grillon, 2008*; *Grillon and Baas, 2003*) and can be implemented in healthy subjects (*Balderston et al., 2017a*, *2017b*; *Cornwell et al., 2007*, *Cornwell et al., 2008*, *2012*; *Lissek et al., 2007*), patients (*Grillon et al., 2009*; *Balderston et al., 2017a*; *Vytal et al., 2016*), and non-human animals (*Davis et al., 2010*). A key feature of prolonged periods of threat of shock is that they induce a stable increase in anxiety that can be probed at random intervals using the acoustic startle reflex

**eLife digest** Anxiety disorders affect around one in five Americans, and in many cases people experience anxiety so intensely that they have difficulties performing day-to-day activities. To help these people, it is important to understand how anxiety works. Current research suggests that anxiety disorders are caused when the connections in the brain that control our response to threat are either excessively or inappropriately activated. However, it was not clear what causes the anxiety to last for long periods. To better understand this phenomenon, Balderston et al. studied the brains of over 30 volunteers using two types of measurements called magnetoencephalography and fMRI.

In the each experiment, participants experienced periods of threat, where they could receive unpredictable electric shocks. In the first experiment, Balderston et al. measured the brain activity by recording the magnetic fields generated in the brain. In the second experiment, they used fMRI to record changes in the blood flow throughout the brain to measure how the different regions in the brain communicate.

The recordings identified a single part of the brain that increased its activity and changed its communication pattern with the other regions in the brain, when people are anxious. This region in a part of the brain called parietal lobe, is also important for processing attention, which suggests that anxiety might make people also more aware of their surroundings. However, this extra awareness might also make it more difficult for people to concentrate.

Future studies may be able to stimulate this area of the brain through the scalp to potentially reduce anxiety, as the affected area is close to the skull.

(*Grillon, 2008*; *Grillon and Baas, 2003*, *1998*), suggesting that this anxious state is mediated by a fundamental sustained change in the pattern of ongoing brain activity. This is in contrast to more phasic event-related fear responses in typical cued fear conditioning (*Schmitz and Grillon, 2012*; *Grillon, 2008*; *Davis et al., 2010*).

Current neuroscientific models of anxiety are based in part on translational work using Pavlovian fear conditioning (*Pavlov, 1927*; *Fanselow and Poulos, 2005*; *Fullana et al., 2016*). Decades of work in non-human animals has shown that acute fear responding is dependent upon the amygdala (*Kwapis et al., 2009*; *Bailey et al., 1999*; *Parsons et al., 2006*), and functional magnetic resonance imaging (fMRI) during fear conditioning in humans has been used to identify a canonical fear network that includes the amygdala, the dorsomedial prefrontal cortex, the thalamus, and the anterior insula (*Fullana et al., 2016*; *Schultz et al., 2012*; *Cheng et al., 2003*, *2006*). However, much less is known about the network mediating extended periods of elevated state anxiety.

In addition, cognitive scientific research in humans shows that attentional processing is profoundly influenced by both state (*Vytal et al., 2013*, *2012*; *Patel et al., 2016*; *Shackman et al., 2006*) and trait anxiety (*Derakshan et al., 2009*; *Eysenck et al., 2007*), suggesting that multiple neural systems are affected by anxiety. Although there have been some studies investigating the neural systems that mediate anxiety, these studies often depend on an *a priori* focus that is centered on the regions of the canonical fear network, and typically rely on *a priori* methods to increase statistical sensitivity in these regions such as lowered statistical thresholds (*Robinson et al., 2013a*; *Mobbs et al., 2010*; *Hooker et al., 2006*; *Tabbert et al., 2010*), region of interest analyses (*Balderston et al., 2015*, *2014*, *2013*), and seed-based functional connectivity (*Schultz et al., 2012*; *Vytal et al., 2014*; *Gold et al., 2015*). Importantly, the increased sensitivity gained by using these statistical methods comes at the cost of assessing anxiety-related changes in regions not identified *a priori*, thus resulting in a possible under-reporting of anxiety-related changes in other areas of the brain, such as regions important for attentional processing. Therefore, the purpose of this study was to use exploratory analytical methods to identify the most prominent activity/connectivity changes induced by the threat of shock paradigm.

Toward this aim, we collected data from two complimentary imaging modalities, fMRI and magnetoencephalography (MEG) during a threat of shock paradigm. In both MEG and fMRI experiments, subjects underwent alternating blocks of safety and threat, and rated their anxiety continuously using a centrally located visual analog scale (see *Figure 1*). During the MEG

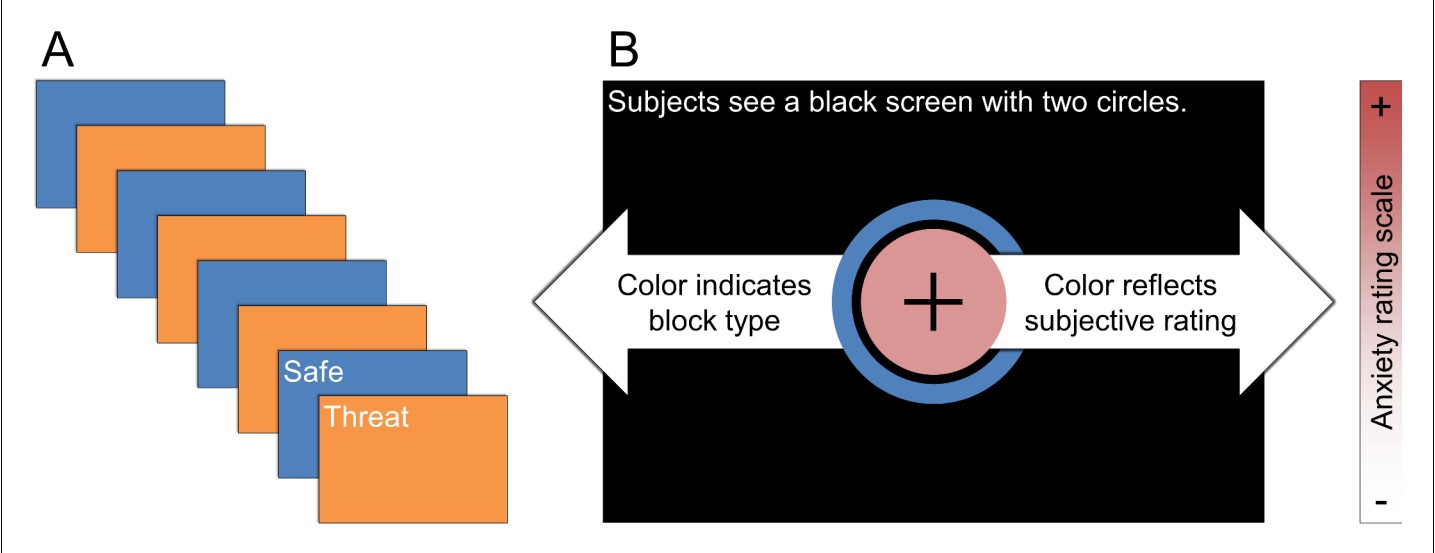

**Figure 1.** Schematic of experimental paradigm. (**A**) Subjects underwent alternating blocks of threat and safety. (**B**) Visual display present on the screen during the experiment. During the experiment subjects saw two circles. The color of the outer circle indicated the block type. The color of the inner circle was controlled by the subject, and reflected the subject's then-current anxiety level.

experiment, subjects also received randomly timed white noise presentations, which served to probe the subject's current anxiety level (via the acoustic startle response) and their ongoing brain activity (via the preceding pattern of neural oscillations).

In both experiments, we used unbiased, data-driven, whole-brain approaches to identify changes in activity (MEG) and connectivity (fMRI) as a function of threat. To assess functional connectivity changes in the fMRI signal, we used the global brain connectivity (GBC) metric, which does not rely on *a priori* seed-selection for the connectivity analysis. To assess ongoing patterns of activity in the MEG study, we evaluated changes in oscillatory power during the 2 s prior to the startle probes as a function of threat. According to the translational approach, one might predict that the most prominent changes in spontaneous neural activity and connectivity would emerge in regions of the canonical fear network (*Fanselow and Poulos, 2005*; *Kim et al., 2011*). However, given that impaired attentional control is a key feature of clinical anxiety (*Derakshan et al., 2009*; *Eysenck et al., 2007*), and that threat of shock has been repeatedly shown to impact performance on tasks that require attention control (*Vytal et al., 2013*, *2012*; *Patel et al., 2016*; *Shackman et al., 2006*), one might also expect that the most prominent changes would emerge within regions of the frontoparietal attention network (*Ptak, 2012*; *Posner, 2012*; *Petersen and Posner, 2012*).

## Results

### Threat increases subjective and physiological measures of anxiety

We began by assessing the ability of our threat manipulation to induce a sustained state of anxiety. Results from the psychological questionnaires, and the subjective rating scales can be found in *Table 1*. Consistent with the online ratings, subjects during both experiments reported more anxiety (MEG: $t(26) = 8.65$; $p<0.001$; fMRI: $t(24) = 13.98$; $p<0.001$) and fear (MEG: $t(26) = 8.03$; $p<0.001$; fMRI: $t(24) = 9.15$; $p<0.001$) during the threat blocks than during the safe blocks. In addition, two sample t-tests did not reveal any significant differences between experiments for either the psychological questionnaires, or the affective rating scales (all ps > 0.05). For the MEG study, we analyzed both the acoustic startle reflex and the online self-reported anxiety ratings. Because startle probes could not be presented during the MRI study, we relied only on the ratings.

For each startle probe in the MEG study, we extracted the subject's startle magnitude, and anxiety rating recorded just prior to the startle probe. Both the startle magnitude and anxiety ratings

**Table 1.** Individual differences for MEG (N = 28) and MRI (N = 25) experiments.

| Measure | MEG | MRI |
| --- | --- | --- |
| *STAI* | | |
| State | 26.04 (1.37) | 23 (0.9) |
| Trait | 27.12 (0.93) | 28.18 (1.27) |
| ASI | 11.59 (1.21) | 8.64 (1.18) |
| BAI | 1.37 (0.42) | 0.58 (0.26) |
| BDI | 0.89 (0.32) | 0.42 (0.19) |
| *Shock* | | |
| Intensity (mA) | 5.66 (0.66) | 6.91 (1.01) |
| Rating | 8.51 (0.2) | 9.09 (0.19) |
| *Anxiety* | | |
| Pre | 2.04 (0.27) | 1.98 (0.25) |
| Safe | 2.47 (0.31) | 1.76 (0.21) |
| Threat | 5.41 (0.37) | 5.97 (0.39) |
| *Fear* | | |
| Pre | 1.41 (0.15) | 1.5 (0.23) |
| Safe | 1.84 (0.27) | 1.27 (0.12) |
| Threat | 4.44 (0.39) | 4.7 (0.42) |

Note: Numbers reflect the mean and standard deviation of the results [M (SD)].

were normalized and converted to T-scores (*Blumenthal et al., 2005*) within subjects. These values were then averaged across trials and submitted to a paired-sample t-test (Safe vs. Threat). Both ratings (See *Figure 2A* and *Figure 2—source data 1*; t(27) = 10.03; p<0.001) and startle (See *Figure 2B* and *Figure 2—source data 1*; t(27) = 4.65; p<0.001) indicated greater anxiety during the threat blocks compared to the safe blocks. Next, we created Threat > Safe difference scores for both startle (anxiety potentiated startle; APS) and the online ratings, and correlated the values across subjects. Ratings within the MEG study were significantly correlated with startle across subjects (See *Figure 2D* and *Figure 2—source data 1*; r(26) = 0.61; p=0.001).

During the MRI study, we averaged the ratings across time in the threat and safe blocks, and converted these values to T-scores. As with the MEG study, these values were submitted to a paired-sample t-test (Safe vs. Threat), and indicated more anxiety during the threat blocks than the safe blocks (See *Figure 2C* and *Figure 2—source data 1*; t(24) = 23.06; p<0.001). We also created Threat > Safe difference scores for these values, and correlated these difference scores with startle (recorded during the MEG study) in the subjects who participated in both sessions. There was a non-significant small positive correlation between startle and rating during the MRI session (see *Figure 2D* and *Figure 2—source data 1*; r(16) = 0.24; p=0.344).

## Threat increases whole brain GBC

Many threat of shock studies have used seed-based functional connectivity analyses to identify changes in emotional processing centers in the brain (*Vytal et al., 2014*; *Gold et al., 2015*; *Satterthwaite et al., 2011*; *Prater et al., 2013*; *Hrybouski et al., 2016*; *Birn et al., 2014*; *Cha et al., 2014*; *Heitmann et al., 2016*); however, seed-based functional connectivity methods suffer from bias because they require the experimenter to select a seed region ahead of time, while ignoring all other possible seed regions. To address this limitation, researchers have developed complementary data-driven functional connectivity metrics such as GBC, which do not rely on *a priori* seed-selection for the connectivity analysis. By assessing the connectivity between each voxel and every other voxel, this analysis allows the user to identify the most connected regions of the brain (*Cole et al., 2010*), as well as the seed regions where connectivity impacts behavior across subjects (*Cole et al., 2012*; *Gotts et al., 2012*). By identifying regions that show the largest changes in GBC

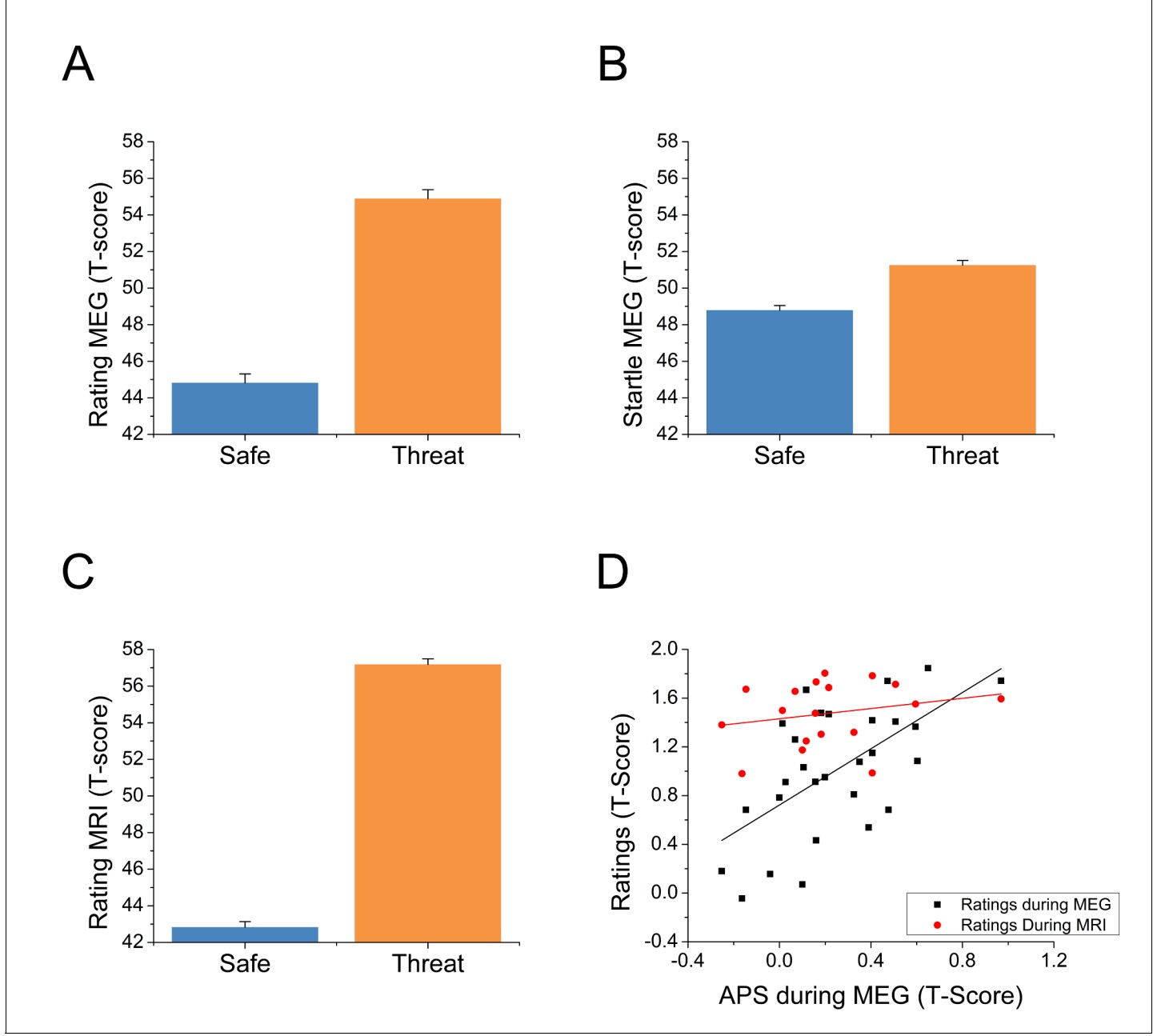

**Figure 2.** Behavioral results from both experiments. (**A**) Anxiety ratings during the MEG study. (**B**) Startle magnitude during the MEG study. (**C**) Anxiety ratings during the fMRI study. Bars represent the mean ± within-subject SEM (*Cousineau, 2005*). (**D**) Correlations between anxiety potentiated startle (APS) and differential anxiety ratings. The black squares represent the correlation between APS and ratings during the MEG session. The red dots represent the correlation between APS during the MEG study and anxiety ratings during the fMRI study in the subset of subjects who participated in both studies.

The following source data is available for figure 2:

**Source data 1.** Source data for all graphs in *Figure 2*.

during periods of threat vs. periods of safety, it is possible to identify hubs that contribute most prominently to the sustained anxious state during threat of shock.

We collected whole brain multi-echo echo-planar imaging data, and used the echo time-dependent independent components analysis to remove sources of noise unrelated to the blood

oxygenation level dependent (BOLD) response from the timeseries (*Kundu et al., 2012*; *Evans et al., 2015*). Subjects were exposed to alternating 2-min blocks of safety and threat, without startle probes. Given that this design lacked external timing information (i.e. external stimulus presentations), we examined changes in functional connectivity as a function of block type. We opted for a data-driven GBC approach where the connectivity of every voxel was assessed. We first computed GBC maps independently for the safe and threat conditions by correlating each voxel's timecourse with every other voxel's timecourse, applying the Fisher's Z transformation, and averaging across these correlation maps (See *Figure 3A,B* and *Figure 3—source data 1*). As a first pass, we averaged across all voxels to obtain the whole brain GBC for safe and threat. We then conducted a (Safe vs. Threat) paired-sample t-test on these values, and found significantly more GBC for threat blocks than for safe blocks (see *Figure 3C* and *Figure 3—source data 1*; t(24) = 2.13; p=0.044).

## Threat increases voxelwise GBC in the intraparietal sulcus (IPS)

To follow-up the whole brain analysis, we conducted a voxelwise analysis of GBC. Using the same GBC maps created above, we conducted a voxelwise (Safe vs. Threat) paired-sample t-test. We used Monte Carlo simulations to estimate a null distribution for statistical testing, and used a cluster-based method based on this null distribution to correct for multiple comparisons. We found a significant increase in GBC in the threat blocks compared to the safe blocks in three clusters (see *Table 2*). The largest cluster was in the right angular gyrus. The two remaining clusters were found bilaterally in the IPS (see *Figure 4A* and *Figure 4—source data 1*). In all three clusters, we found significantly higher GBC for threat blocks than for safe blocks (see *Figure 4B* and *Figure 4—source data 1*). To determine whether these differences were affected by the delivery of the shock, or differences in motion across blocks, we repeated the threat vs. safe analysis at the cluster level after censoring the 10 TRs following shock delivery, and an equivalent number of safe TRs closely matched for motion. In addition, we covaried out any remaining differences in motion using an analysis of covariance. Using this approach, we still found a significant effect of threat on GBC in all three regions (Right angular gyrus: f(1, 23)=4.71, p=0.04; Right IPS: f(1, 23)=7.22, p=0.01; Left IPS: f(1, 23)=5.39, p=0.03), suggesting that our initial findings were not due to differences in motion, censoring, or residual neural activity evoked by the shock.

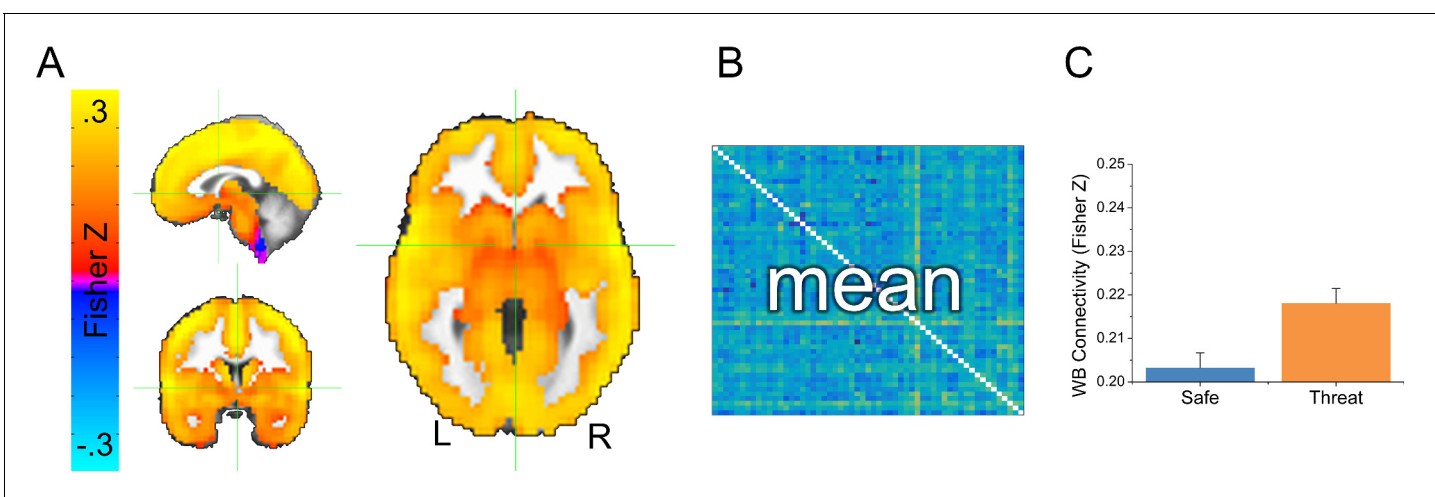

**Figure 3.** Overview of global brain connectivity (GBC) measure. (**A**) Map showing average GBC across all safe and threat TRs. (**B**) Cartoon schematic of a correlation matrix. The 43204 voxel x 43204 voxel cross correlation matrix was calculated separately for each subject and each condition. Correlations were averaged across rows for the entire grey matter mask, to create a single map reflecting the average correlation between each voxel and all other voxels in the mask. (**C**) Graph representing the mean GBC following the Fisher's Z transformation for safe and threat averaged across the entire grey matter mask. Bars represent the mean ± within-subject SEM (*Cousineau, 2005*).

The following source data is available for figure 3:

**Source data 1.** Source data for graph in *Figure 3C*.

**Table 2.** Results from voxelwise GBC analysis.

| Label | Volume | t-value | Peak activation (LPI) | | |
|---|---|---|---|---|---|
| | | | x | y | z |
| Right Angular Gyrus | 158 | 3.45 | 48 | −51 | 27 |
| Right Intraparietal Sulcus | 83 | 3.42 | 21 | −60 | 66 |
| Left Intraparietal Sulcus | 81 | 3.6 | −18 | −63 | 66 |

## Threat increases connectivity within an IPS-centered attentional network

Given that the IPS emerged as a hub differentiating global connectivity in threat compared to safe, we used this region as a seed-region to identify changes in functional connectivity during the threat vs. safe blocks. We understand this follow-up analysis could be interpreted as circular. According to this perspective, the seed-based analysis is not independent from the GBC analysis, which was used to identify the seed. However, the purpose of the follow-up analysis (i.e. probing functional connectivity using clusters identified from the global connectivity analysis) is to limit the interpretations of the global connectivity results to those supported by seed-based functional connectivity results, which has been done in previous group-level GBC studies (e.g. [*Gotts et al., 2012*]).

We extracted the timecourse of activity averaged across the voxels in the bilateral IPS functional ROIs, and correlated this timecourse with the timecourse of activity across all voxels in the brain, independently for safe and threat, and applied the Fisher's Z transformation to the resulting correlation coefficients. We conducted a voxelwise (Safe vs. Threat) paired-sample t-test on the resulting maps. We used Monte Carlo simulations and cluster thresholding to correct for multiple comparisons. Consistent with the GBC results, we found an increase in connectivity in threat blocks

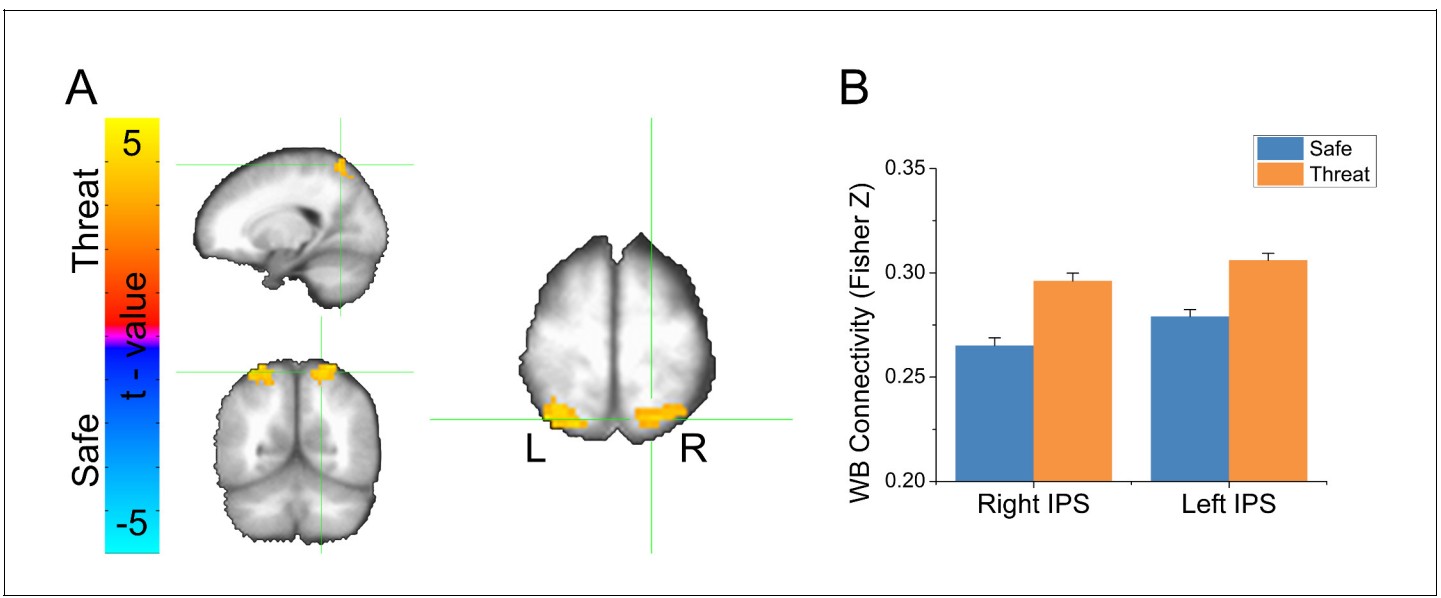

**Figure 4.** Results from voxelwise global brain connectivity (GBC) analysis. (**A**) Statistical map showing results from a threat vs. safe paired-sample t-test. (**B**) Graph representing average GBC values after applying the Fisher's Z transformation for clusters shown in panel A. Bars represent the mean ± within-subject SEM (*Cousineau, 2005*).
The following source data is available for figure 4:

**Source data 1.** Source data for graph in *Figure 4B*.

compared to safe blocks, in several regions of the frontoparietal attention network (See *Table 3*, *Figure 5*, and *Figure 5—source data 1*).

## Alpha oscillations dominate neuromagnetic recordings at rest

Next, we characterized the pattern of activity in the MEG study. It is well established that spontaneous neural activity at rest is dominated by oscillations in the alpha (8–12 Hz) range (*de Munck et al., 2008*; *Sadaghiani et al., 2010*; *Mo et al., 2013*; *Mayhew et al., 2013*), which are most prominent when the subject is in an alert state of restful relaxation (*Doufesh et al., 2014*; *Kim et al., 2014*; *Khalsa et al., 2015*; *Lagopoulos et al., 2009*), and that alpha asymmetries can reflect differences in arousal across subjects (*Nitschke et al., 1999*). Theoretical models of alpha function suggest that alpha oscillations are generated by coherent activity in local inhibitory interneurons (*Klimesch et al., 2007*) and that decreases in alpha power reflects increases in cortical excitability (*Klimesch et al., 2007*; *Lange et al., 2013*). Consistent with these theories, studies collecting simultaneous measures of electroencephalography (EEG) and fMRI have shown that alpha power is negatively correlated with functional connectivity (*Laufs et al., 2003*; *Scheeringa et al., 2012*; *Chen et al., 2008*; *Chang et al., 2013*).

We extracted and cleaned the 2 s of data prior to each startle probe, avoiding contamination by blink artifacts. Because the timing of the startle probes was random, the pre-stimulus recording reflected random sampling across the sustained threat period. Therefore, we collapsed across the 2 s interval and examined oscillatory activity. First, we transformed the values into the frequency domain using a Fast Fourier Transform (FFT) with upper and lower limits of 20 Hz and 1 Hz, respectively. Then, we averaged these values across sensors, trials, and subjects, and examined the spectrogram for peaks. We detected a peak at ~10 Hz, suggesting that alpha oscillations were a prominent feature of these recordings (*Figure 6A* and *Figure 6—source data 1*). We identified the largest local maxima in each subject's spectrogram. In all but four subjects we detected a peak in the alpha frequency band (8 Hz – 12 Hz). The power within a 2 Hz band around these individual alpha frequencies (IAF)s was used in all subsequent analyses (*Figure 6B* and *Figure 6—source data 1*). For subjects without a detectable IAF, power in a narrow band around IAF averaged across subjects was used. Subsequent analyses were performed in both sensor space (*Figure 6C*; black dots) and source space (*Figure 6C*; green dots).

## Threat reduces parietal alpha oscillations

Given that alpha oscillations were the dominant feature in the MEG recordings across all blocks, we examined whether these oscillations differed as a function of threat. For the sensor space analysis, we averaged IAF across trials within conditions, and then performed a paired-sample t-test (Safe vs. Threat) on the resulting sensor space averages. We used Monte Carlo simulations and cluster thresholding to correct for multiple comparisons. Importantly, we found a large cluster of sensors over the left parietal lobe and a smaller cluster of frontal sensors showing significantly less IAF power in the threat blocks than during the safe blocks (*Figure 7A–B* and *Figure 7—source data 1*).

**Table 3.** Results from voxelwise IPS connectivity analysis.

| Label | Volume | t-value | Peak activation (LPI) | | |
|---|---|---|---|---|---|
| | | | x | y | z |
| Left Thalamus | 342 | 3.92 | -9 | 6 | 12 |
| Right Inferior Parietal Lobule | 208 | 3.67 | 57 | −57 | 39 |
| Left Superior Medial Gyrus | 184 | 3.65 | 3 | 36 | 42 |
| Left Precuneus | 179 | 3.59 | 3 | −69 | 48 |
| Right Middle Frontal Gyrus | 137 | 3.64 | 33 | 15 | 60 |
| Left Angular Gyrus | 113 | 3.51 | −57 | −54 | 30 |
| Left Middle Frontal Gyrus | 96 | 3.69 | −24 | 15 | 60 |
| Left Middle Frontal Gyrus | 90 | 3.48 | −45 | 51 | -3 |

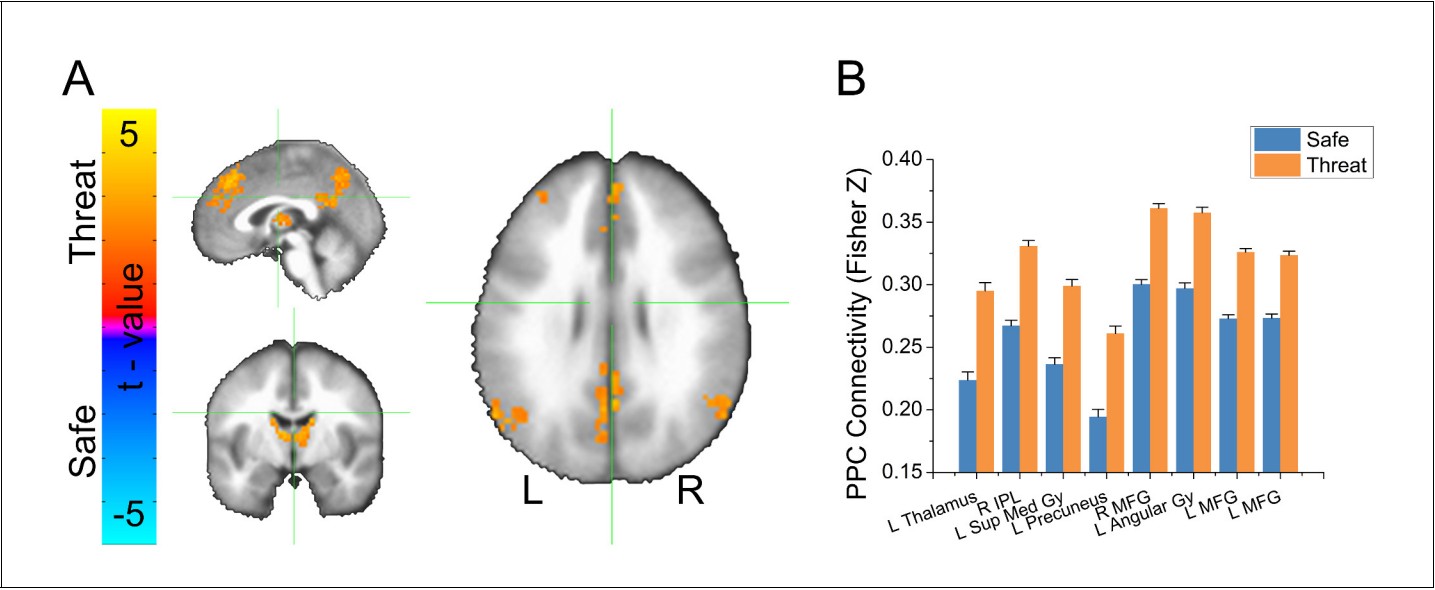

**Figure 5.** Results from bilateral IPS seed-based connectivity analysis. (**A**) Statistical map showing results from a threat vs. safe paired-sample t-test. (**B**) Graph representing average IPS connectivity values for clusters shown in panel A. Bars represent the mean ± within-subject SEM (*Cousineau, 2005*).

The following source data is available for figure 5:

**Source data 1.** Source data for graph in *Figure 5B*.

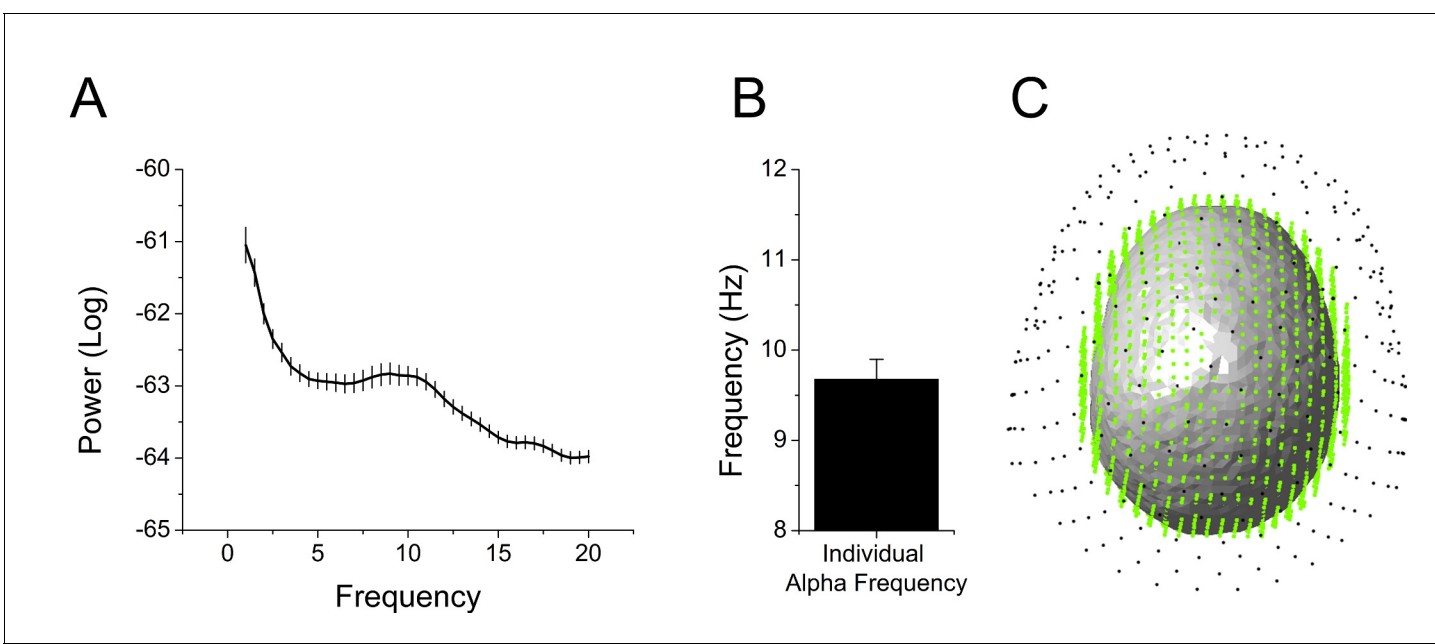

**Figure 6.** Overview of MEG analyses. (**A**) Spectrogram representing power averaged across all subjects and all sensors with peak in the alpha frequency band. (**B**) Graph showing the frequency of peak alpha (individual alpha frequency) averaged across subjects. Bars represent the mean ± SEM. (**C**) Example of single subject alignment with sensors (black dots) source grid (green dots) and headmodel (surface) plotted together.

The following source data is available for figure 6:

**Source data 1.** Source data for graph in *Figure 6A and B*.

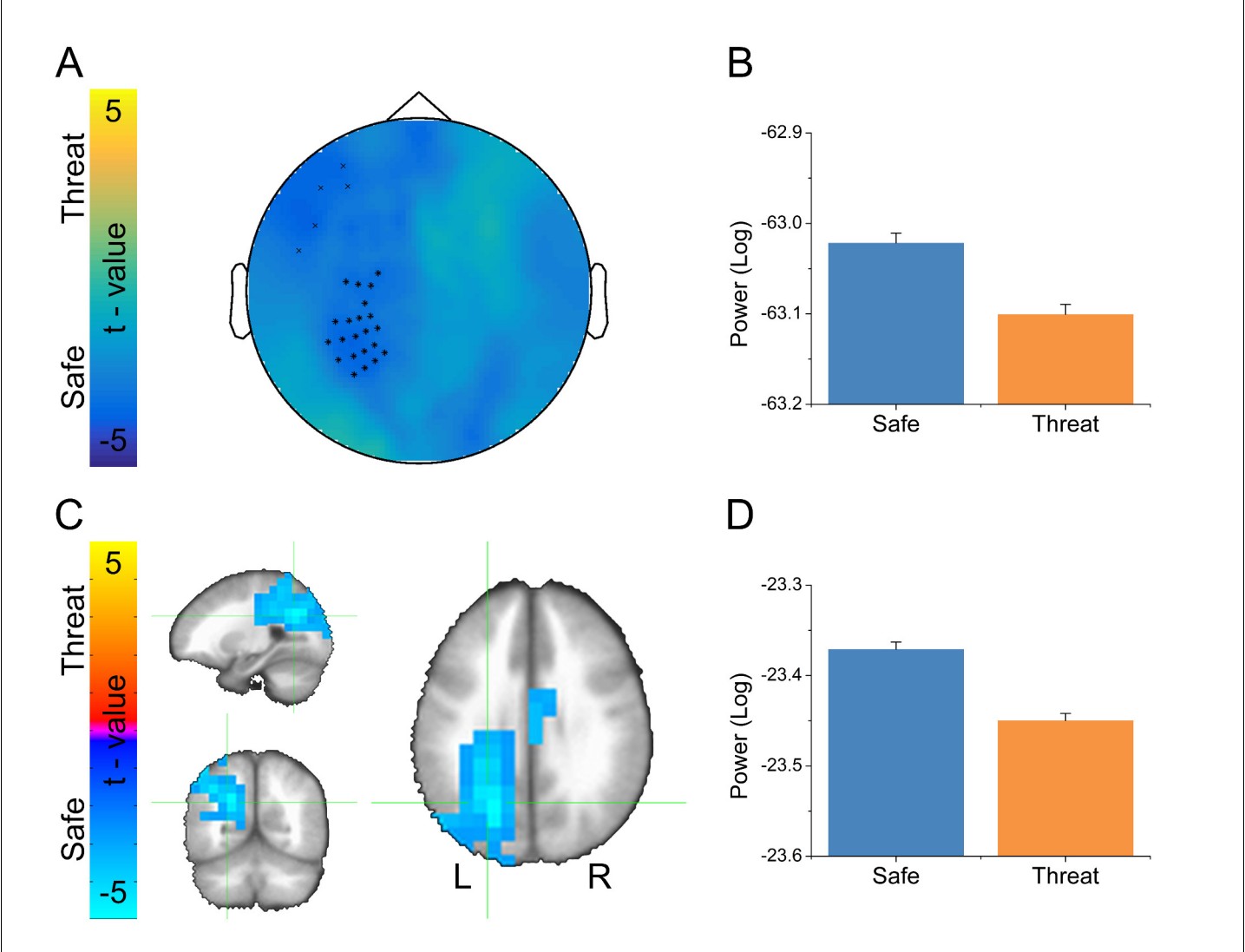

**Figure 7.** Alpha results from threat vs. safe t-test. (**A**) Statistical map in sensor space showing a significant reduction in alpha power. Black symbols represent clusters of sensors showing significant threat vs. safe differences. (**B**) Graph showing average alpha power for safe and threat conditions in the largest cluster of sensors in panel A. (**C**) Statistical map in source space showing a significant reduction in alpha power. (**D**) Graph showing average alpha power for safe and threat conditions in the cluster in panel C. Bars represent the mean ± within-subject SEM (***Cousineau, 2005***).

The following source data is available for figure 7:

**Source data 1.** Source data for graph in ***Figure 7B and D***.

To localize the source of these power differences, we projected these signals into source space using a dynamic imaging of coherent sources (DICS) beamformer. First, we created a common filter using all trials, then we projected the safe and threat trials through the filter independently, to obtain power estimates at the source level for each condition. Finally, we conducted a (Safe vs. Threat) paired-sample t-test on the source space IAF power estimates. As with the sensor space data, we used Monte Carlo simulations and cluster thresholding to correct for multiple comparisons. Consistent with the sensor space results, we found two clusters of voxels showing significantly less IAF power in the threat and safe conditions (***Figure 7C–D*** and ***Figure 7—source data 1***), the larger cluster was located in the left IPS, while the smaller cluster was located in the mid-cingulate gyrus. In addition, both the sensor and the source space results held if we analyze power across the entire alpha frequency band (8 Hz to 12 Hz; See below). Finally, when comparing the MEG results with the

fMRI results, we found that the left IPS cluster in the fMRI data substantially overlaps with the alpha cluster (46/81 voxels; See *Figure 8*).

As with the MRI data, it is important to show that our denoising steps did not affect the comparisons between the safe and threat conditions. Specifically, it is important to show that the number of trials rejected either (1) did not differ across the safe and threat blocks, or (2) did not affect the safe vs. threat comparisons. In the safe condition, there were on average 58.21 ± 5.5 trials, while in the threat condition there were on average 55.32 ± 7.6 trials, which was significantly different across subjects (t(27) = 2.78, p=0.01). Therefore, we decided to determine whether this difference in trial number impacted our results at the censor and source level. Accordingly, we included the difference in trial number across safe and threat blocks as a covariate in an ANCOVA examining the effect of threat on alpha responses. At both the sensor (f(1,26) = 7.48, p=0.01) and at the source (f(1,26) = 17.797, p<0.001), we find that even with the difference in number of trials covaried out, the effects of threat on alpha power is still significant, suggesting that this difference did not significantly impact our results.

## Discussion

The purpose of this study was to use the complementary methods of fMRI and MEG to identify the regions and network hubs that contribute most prominently to sustained anxiety during threat of shock. In both cases, we used unbiased, data-driven, whole-brain measures of activity/connectivity, and identified the IPS as the region most affected by the threat of shock manipulation. Using fMRI

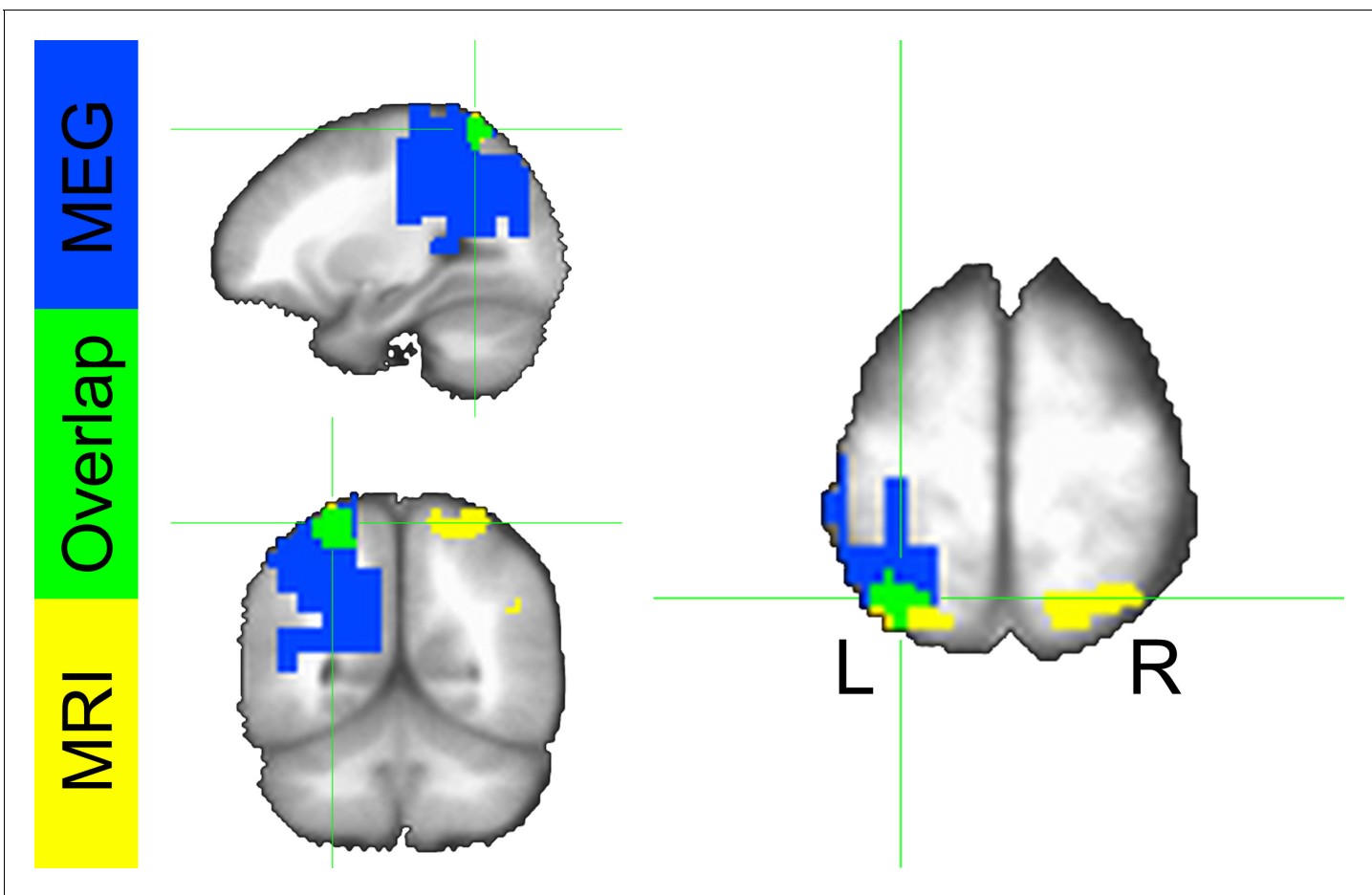

**Figure 8.** Conjunction map from voxelwise fMRI GBC analysis and MEG alpha power differences. Colors represent significant safe vs. threat differences from the fMRI analysis (yellow), MEG analysis (blue), and both analyses (green).

GBC, we found that the IPS increased its connectivity during periods of threat with a set of regions important for attention control, suggesting that this region is a hub in a network important for attentional processing during threat (*Du et al., 2012*; *Kincade et al., 2005*; *Rushworth et al., 1997*; *Bucci, 2009*; *Corbetta and Shulman, 2002*). Using MEG, we found that the magnitude of spontaneous alpha oscillations in the same region decreased during periods of threat, suggesting that threat increases IPS cortical excitability (*Klimesch et al., 2007*). Together, these results suggest that threat of shock facilitates attentional processing mediated by increased excitability and connectivity of the IPS (*Du et al., 2012*; *Kincade et al., 2005*; *Rushworth et al., 1997*; *Bucci, 2009*). This enhanced attentional processing may reflect a state of hypervigilance, consistent with the findings that elevated state anxiety biases the attentional system (*Waters et al., 2014*; *Amir et al., 2009*; *Lapointe et al., 2013*). In addition, this hypervigilance may provide a mechanistic explanation for why threat facilitates performance on tasks that require an external focus of attention but impairs performance on tasks that require an internal focus of attention (*Vytal et al., 2016*, *2013*; *Patel et al., 2016*; *Torrisi et al., 2016*; *Grillon et al., 2016*; *Robinson et al., 2011*; *Balderston et al., 2016*).

## Functional connectivity

Previous studies exploring the relationship between threat and functional connectivity focused on how threat affected connectivity within networks centered on seeds placed in emotional processing regions (*Vytal et al., 2014*; *McMenamin et al., 2014*; *McMenamin and Pessoa, 2015*). However, in this study because we were specifically interested in identifying the regions of the brain that contributed most prominently to anxiety, we chose to forego an *a priori* seed-based approach, and employ data-driven connectivity measures that do not rely on choosing a seed (*Cole et al., 2010*; *Calhoun et al., 2009*). To start, we used GBC as a method to determine whether connectivity *en masse* changed with threat, and to identify where in the brain the largest changes occurred (*Cole et al., 2010*, *2012*; *Chu et al., 2011*). We found the largest changes to occur bilaterally in the IPS, where connectivity with the rest of the brain increased as a function of threat. Then, using these regions as a seed, we found that they supported an increase in connectivity within a set of regions important for executive control (*Posner, 2012*; *Corbetta and Shulman, 2002*). These results suggest that the IPS is a critical connectivity hub in the network of regions that contribute most prominently to the sustained elevation of state anxiety induced by the threat of shock paradigm (*Cole et al., 2010*, *Cole et al., 2014*). Although our GBC approach did not reveal any hubs within the amygdala, or in other regions typically associated with emotional processing (*Simmons et al., 2006*; *Mechias et al., 2010*; *Etkin and Wager, 2007*), we do see an increase in connectivity between the IPS and the thalamus and dorsomedial prefrontal cortex, two regions known to be part of the canonical fear network.

In addition to the IPS, we found changes in GBC as a function of threat in the right angular gyrus. Although it is currently unclear how the angular gyrus might contribute to threat processing, there is work suggesting that this region may be a key site for multisensory integration (*Seghier, 2013*). One hypothesis is that enhanced GBC in this region may reflect a heightened awareness of the current context. However, this is a post hoc explanation that should be explored in future studies.

An obvious parallel to the fMRI connectivity analysis would be to conduct a similar whole brain connectivity analysis of the MEG data. Not only would corroborating evidence from an independent imaging modality strengthen the fMRI connectivity results, but results from MEG specifically would allow for a frequency-specific analysis of the effects of threat on functional connectivity (*Hillebrand et al., 2012*; *Brookes et al., 2011*). However, the current study was not optimized to reliably detect differences in MEG connectivity. In the current study, we included the white noise probes as a way to obtain a quantitative measure of anxiety throughout the safe and threat blocks (*Grillon et al., 1999*). These white noise probes trigger the acoustic startle reflex, which varies as a function of an individual's current level of anxiety (*Grillon, 2008*). Unfortunately, these reflexive blinks also inject an artifact into the MEG signal, and because the magnitude of these blinks differs across conditions, the blink artifact also differs across conditions. Although adaptive beamforming can theoretically remove the artifact induced by the blink response (*Van Veen et al., 1997*), the only way to ensure that the blink artifact does not differentially affect the MEG signal is to remove the contaminated time periods from the analysis, or remove the startle probes from the design at the outset. In the current study, we chose to address this limitation by extracting two-second time

windows prior to each startle presentation to minimize the effect of the blink artifact on our power estimates. However, it has been shown that reliability of the MEG connectivity estimates increases as the duration of the recording increases, and durations of ~10 min or greater may be needed to maximize reliability (*Liuzzi et al., 2016*). Therefore, using such short intervals did not allow for the ability to obtain reliable estimates of MEG connectivity. Future studies should be conducted to address this limitation. In addition, it will be important to use appropriate correction methods to account for signal leakage between source space regions (*Colclough et al., 2015*), and verify that the resulting connectivity estimates match previously published work (i.e. strong alpha connectivity in occipital regions and strong beta connectivity linking the parietal cortex with other frontal, temporal, and occipital regions; [*Hunt et al., 2016*]).

## fMRI and alpha

In addition to the increases in functional connectivity measured with fMRI, we also found decreases in alpha as a function of threat of shock in the same parietal region using MEG. Although the focus of the paper was on alpha, our initial approach (described in the introduction) was to examine all frequency bands independently (other bands not shown). The focus on alpha emerged out of the observations that, (1) alpha was the strongest signal in the recordings, (2) alpha showed the largest power changes as a function of threat, (3) alpha was the only frequency band that showed consistent results at both the sensor and the source level, and (4) the source space results aligned nicely with the corresponding fMRI GBC data.

These results are also consistent with several previous studies using simultaneous recordings of fMRI and EEG (*Sadaghiani et al., 2010*; *Mo et al., 2013*; *Mayhew et al., 2013*; *Scheeringa et al., 2012*; *Chang et al., 2013*; *Wu et al., 2010*; *Walz et al., 2015*; *Scheeringa et al., 2011*). For instance, functional connectivity at the whole brain level (*Chang et al., 2013*), and within the visual system (*Scheeringa et al., 2012*) is negatively correlated with posterior alpha power. In addition, alpha power is negatively correlated with activity with dorsal attention network (*Sadaghiani et al., 2010*), and positively correlated with activity in the default mode network (*Mo et al., 2013*). Finally, intrinsic connectivity within networks is negatively correlated with alpha during eyes open vs. eyes closed resting state studies (*Wu et al., 2010*), and both the phase and power of pre-stimulus alpha affects event-related BOLD responses in sensory regions (*Mayhew et al., 2013*; *Walz et al., 2015*; *Scheeringa et al., 2011*). Together these results suggest that the increase in IPS connectivity and decrease in IPS alpha power may reflect a common process.

## Alpha oscillations

We also show that threat reduces the power of resting state alpha oscillations. These oscillations are thought to reflect cortical inhibition, driven by inhibitory interneurons, and triggered by top-down modulatory control (*Klimesch et al., 2007*). According to this view, our current findings may be driven by a release from top-down inhibition, resulting in a net increase in cortical excitability. Importantly, reductions in alpha power (i.e. increases in excitability) are associated with enhanced sensory and motor processing (*Cornwell et al., 2007*; *Baas et al., 2006*). For instance, reduced pre-stimulus alpha is associated with enhanced visual and illusory visual perception (*Lange et al., 2013*), and increased pre-stimulus alpha is associated with reduced transcranial magnetic stimulation-induced motor-evoked potentials (*Klimesch et al., 2007*). Accordingly, alpha reduction (i.e. cortical excitation) may provide a mechanistic explanation for one of the most commonly reported symptoms of elevated state anxiety, namely anxiety-potentiated startle (*Schmitz and Grillon, 2012*; *Grillon and Ameli, 1998*; *Grillon et al., 1991*). That is, threat of shock potentiates the startle reflex by increasing cortical excitability.

Cognitively, alpha oscillations are thought to play a key role in selective attentional processes (*Klimesch, 2012*), such that increases in alpha typically reflect inhibition or filtering of information that is to-be-ignored (*Bonnefond and Jensen, 2013*; *Kelly et al., 2006*; *Händel et al., 2011*). For instance, alpha oscillations have been shown to filter out noise during distracting listening conditions (*Strauß et al., 2014*). Similarly, increases in pre-stimulus alpha to predictable stimuli are lateralized to to-be-ignored locations (*Horschig et al., 2014*; *Rihs et al., 2009*). In addition, alpha power increases during the maintenance of items in working memory (WM) (*Klimesch et al., 2007*; *Meyer et al., 2013*), and this increase in alpha serves to protect these items from distractors

(*Bonnefond and Jensen, 2012*; *Manza et al., 2014*). Taken together, these results suggest that the decreases in alpha power observed in our study may reflect greater perceptual sensitivity, consistent with the idea that threat may increase vigilance (*Eilam et al., 2011*). In addition, alpha-gamma coupling is thought to be important for allocating attention to unattended salient stimuli (*Jensen et al., 2012*).

The reductions in alpha power observed here were lateralized to the left hemisphere, leading to a right dominant parietal asymmetry. According to prominent models, left dominant frontal alpha is associated with positive affect and/or approach behavior, while right dominant alpha is associated with negative affect and/or avoidance behavior (*Davidson, 2004*; *Harmon-Jones et al., 2010*). Like the valence model of alpha asymmetry, one prominent model on parietal alpha asymmetry is rooted in the arousal-valence model of emotional processing (*Heller, 1993*). According to this model, right dominant parietal alpha is associated with increased arousal. Consistent with this theory, we find that threat of shock, which increases arousal, also reduces left parietal alpha, resulting in a right dominant profile. However, more research should be conducted to specifically test this hypothesis.

## Cognition/anxiety interactions

Although our results only provide indirect evidence for the hypothesis that threat facilitates orienting, a bias toward hyper-orienting during periods of elevated state anxiety could explain many of the conflicting findings related to the cognition/anxiety interaction. The relationship between cognition and anxiety has been extensively studied (For reviews, see [*Eysenck et al., 2007*; *Robinson et al., 2013b*]). Importantly, threat of shock improves performance on sustained attention tasks (*Torrisi et al., 2016*; *Grillon et al., 2016*; *Robinson et al., 2011*), while impairing performance on working memory tasks (*Vytal et al., 2016*, *2013*; *Patel et al., 2016*; *Balderston et al., 2016*). These tasks can be distinguished from one another based on the locus of attention required for performance. In sustained attention tasks, subjects are constantly monitoring the external environment for odd-ball stimuli, while in working memory tasks, subjects are required to maintain information internally. Our hypothesis is that threat-induced hyper-orienting improves performance on sustained attention tasks by reducing lapses in attention (*Torrisi et al., 2016*). In contrast, we hypothesize that threat-induced hyper-orienting impairs performance on working memory tasks by lowering the threshold for distractors to gain access to WM resources (*Balderston et al., 2016*). Consistent with these hypotheses, high pre-stimulus alpha is associated with better memory performance while low pre-stimulus alpha is associated with better perception performance (*Klimesch et al., 2007*).

## Strengths and limitations

This study had a number of strengths. First, we used the gold standard threat of shock paradigm, which has been extensively applied to manipulate state anxiety within subjects (*Grillon, 2008*; *Grillon and Baas, 2003*, *1998*; *Grillon et al., 1991*, *2007*, *2008*, *2009*; *Balderston et al., 2017a*, *2017b*; *Cornwell et al., 2007*, *Cornwell et al., 2008*, *2012*; *Lissek et al., 2007*). Unlike previous studies, we also collected anxiety ratings throughout the recordings, and found that these ratings were similar across studies and correlated with anxiety-potentiated startle within session. Similarly, retrospective anxiety ratings were similar across studies, suggesting a comparable anxiety induction in both studies. We also collected data from multiple imaging modalities that both support the finding of enhanced IPS processing during periods of threat. In addition, we used the state-of-the-art multi-echo fMRI acquisition, and accompanying echo time-dependent independent components analysis to eliminate non-BOLD sources of noise from our fMRI data.

Although there were a number of strengths to our study, there were also several limitations. First, as mentioned above, our MEG study was not optimally designed to reliably detect differences in connectivity, which would have been an obvious parallel to the fMRI connectivity analysis. Second, because we had a relatively small sample size, this study was not optimally designed to study how individual variability in anxiety affected IPS activity/connectivity. Third, our initial goal was to collect both the fMRI and the MEG data in all subjects, but a number of subjects could not participate in both, making it difficult to compare the results at the single subject level. Fourth, given our interest in the relationship between alpha and connectivity, it might have been better to collect simultaneous fMRI and EEG. However, because of our interest in the startle data, it was important to present white noise probes in at least one of the experiments, and our MRI scanner does not currently have

that capability. To overcome that limitation, we collected startle data in the MEG study, which we used to validate the continuous anxiety ratings, which were collected during both studies. Finally, another limitation with the MEG data is that we do not have a measure of within-block movement, making it difficult to determine whether motion differed between the safe and threat blocks; however, it should be noted that any trials contaminated by movement (i.e. muscle) artifact were removed.

## Conclusions

Current translational neuroscientific models of anxiety focus on regions of the canonical fear network as anatomical hubs for anxiety. However, support for these models in humans is often based on studies that focus on these regions *a priori* (i.e. seed-based functional connectivity), at the expense of rigorous, unbiased, whole-brain approaches. In this work we conduct two separate experiments, using complimentary imaging modalities (fMRI and MEG) to assess the effect of elevated state anxiety on functional connectivity and cortical excitability across the entire brain. In these studies, we identify effects in a common region, the IPS, which is a key node in the frontoparietal attention network. These results suggest that threat enhances processing in this region, possibly facilitating attentional processing, leading to increased vigilance. However, these results have broader implications for future research and treatment. First, our results suggest that elevated anxiety in humans is primarily a cognitive state that is not fully captured by the current translational models. Second, because the most prominent region contributing to elevated state anxiety is cortical, it may be possible to target this region with neuromodulatory methods and reduce anxiety.

# Materials and methods

## Participants

Forty-two (28 female; age: M = 28.45, SD = 6.23) healthy volunteers from the Washington DC metropolitan area were recruited to participate in the current study. Of these, 32 (18 female; age: M = 27.82, SD = 5.11) participated in the magnetoencephalography (MEG) study, and 30 (19 female; age: M = 28.52, SD = 6.75) participated in the functional magnetic resonance imaging (fMRI) study. Among the subjects included in the final analysis, 18 (13 female; age: M = 27.67, SD = 4.76) participated in both. For the MRI study, two subjects were removed due to technical malfunction with the scanner/data, three subjects were removed because their anxiety ratings were two standard deviations below the mean. For the MEG study, three subjects were removed from the final analysis for sleeping, moving, or not paying attention to the task. One subject withdrew in the middle of the recording. The final sample included 38 total subjects (MEG N = 28; MRI N = 25): 18 subjects participated in both experiments, 10 participated only in the MEG experiment, and seven participated only in the MRI experiment. Although no explicit power analysis was conducted prior to the experiment, we chose a sample size of approximately 25 participants, to ensure enough power to detect a behavioral threat of shock effect based on previous studies (*Schmitz and Grillon, 2012*; *Balderston et al., 2017b*, *2016*).

Following an initial telephone screen, participants visited the National Institutes of Health Clinical Center for a comprehensive screening by a clinician. Inclusion criteria for healthy volunteers were: (1) no current Axis I psychiatric disorder as assessed by SCID-I/NP (*First et al., 2012*), (2) no first-degree relative with a known psychotic disorder, (3) no interfering acute or chronic medical condition, (4) no brain abnormality on MRI as assessed by a licensed radiologist, (5) negative urine drug screen, and (6) right-handedness. All participants gave written informed consent approved by the National Institute of Mental Health (NIMH) Combined Neuroscience Institutional Review Board and received compensation for participating.

## Stimulus presentation

Presentation software package (version 14.6, Neurobehavioral Systems, Berkeley, CA) was used to present the stimuli via projection systems in both the MEG (front) and MRI (rear projection) studies.

## Shock

A 100 ms, 200 Hz train of stimulation was administered via a Digitimer constant current stimulator (DS7A; Digitimer, Letchworth Garden City, UK). The transistor–transistor logic (TTL) pulse train used to trigger the train of stimulation was generated via a Grass stimulator (SD9, Warwick, RI). In the MEG experiment, this shock was delivered to the subjects' right wrist via two 8 mm Ag/AgCl surface cup electrodes (EL258-RT, Goleta, CA), filled with electrolyte gel (GEL100, Goleta, CA). In the MRI experiment, the same shock was delivered to the subjects' right wrist via two 11 mm Ag/AgCl MRI-safe disposable sticker electrodes (EL508, Goleta, CA). The intensity of the shock could range from 0 mA to 100 mA and was calibrated prior to the experiment to a level that the subject rated as 'uncomfortable but not painful'.

## Acoustic startle stimulus

During the MEG study, subjects were exposed to several presentation of an acoustic white noise, to trigger an acoustic startle reflex used to assess anxiety. In order to avoid artifact due to the magnets and moving metal in traditional headphone drivers, we engineered a custom pneumatic system for generating white noise with air pressure. First we used a 3D printer to create plastic over-the-ear air vortex generators (vortices). Then, we attached these to tubes connected to a solenoid, which was connected to an air tank. When triggered by a TTL pulse, the solenoid allowed air from the tank to pass through tubes to the air vortex generators. In doing so, the air generated a white noise with an intensity proportional to the air pressure released. Therefore, we calibrated the volume of the white noise to 103 dB by adjusting the pressure on the air tank regulator, and testing the intensity with a sound pressure level meter.

## Response devices

During the experiment, subjects had continuous access to an online visual analogue scale that they could use to continuously update their anxiety rating during the task. Subjects controlled this scale using the response device provided for each experiment. In the MEG experiment, subjects used a custom-built fiber optic joystick. In the MRI experiment, subjects used a 4-button fiber optic response device (Current Designs, Philadelphia, PA).

## Eyeblink reflex

The acoustic startle reflex was measured during the MEG study via electromyographic (EMG) activity of the eyeblink reflex recorded via 2 8 mm Ag/AgCl surface cup electrodes (EL258-RT, Goleta, CA), filled with electrolyte gel (GEL100, Goleta, CA) placed below the right eye over the *orbicularis oculi* muscle (*Blumenthal et al., 2005*). EMG was recorded at 600 Hz via the MEG system amplifier and analyzed using a custom MATLAB script (see *Source code 1*). The EMG signal was extracted from the recordings, bandpass filtered (30–500 Hz), rectified, and smoothed with a 20 ms time constant. The peak startle/eyeblink magnitude was determined during the 20–100 ms after the onset of the noise presentation. The peaks were then transformed to *z*-scores and converted to *t*-scores within-subjects to reduce large inter-individual differences in the overall magnitude of the startle reflex (*Blumenthal et al., 2005*), and to facilitate comparison with the online anxiety ratings.

## Online ratings

Subjects reported their anxiety level continuously throughout the experiment using the response device. They also received continuous feedback via a colored circle in the center of the screen, updated at 60 Hz (A colored circle, as opposed to a moving cursor [*Schultz et al., 2012*; *Balderston and Helmstetter, 2010*], was chosen to minimize eye movements in the MEG study). As they updated their rating it updated the color of the circle, which ranged from white to red with 256 possible intermediate hues. Values representing these hues were used to numerically represent the subjects' current anxiety level. During the MEG study, these ratings were sampled just prior to each startle probe for comparison with the startle magnitudes. Because startle probes were not presented during the MRI study, online ratings were sampled once per repetition (TR). The values were then transformed to *z*-scores and converted to *t*-scores within-subjects to facilitate comparison with the startle responses.

## Procedure

### MEG

On the day of the appointment, the subject arrived at the NIH clinical center, and completed the informed consent form, and pre-experiment questionnaire packet in the waiting room. Next, subjects were escorted to the MEG suite and given the instructions for the task. Afterward, the subject was prepped to enter the magnetically shielded room (MSR). Electrodes to deliver the shock and measure heartbeats/eyeblinks were attached and secured to the subject. In addition, head position indicator (HPI) coils were attached to the subject. One each was attached 1.5 cm anterior to the left and right tragi, and one was attached 1.5 cm superior to the nasion. Finally, the vortices were affixed to the subject's hairnet, and secured over the ears. Once setup was completed, the subject was escorted into the MSR, given the shock workup procedure (*Balderston et al., 2017a*, *2017b*), and raised into place. Recording began with a 2-minresting run to ensure that there were no visual artifacts (breathing, jaw movement, etc.) in the data. After the resting run, the subject was asked the pre-experiment affective rating scales over the intercom. Once complete, the subject began the experiment.

During the experiment, the subject viewed two concentric circles (See *Figure 1*). The color of the outer circle indicated the type of block (orange = threat; blue = safe). The color of the inner circle indicated the subject's current level of anxiety. The subject was instructed to keep the color scale updated continuously throughout the experiment with their current level of anxiety. They were instructed that red meant that they were extremely anxious, while white meant that they were not anxious at all. They were also instructed that they could adjust the color to any shade between red and white, depending on their anxiety level.

The experiment consisted of four runs, each of which were ~6 min long, determined dynamically by the length of the trials. Each run began with four presentations of the startle probes to facilitate habituation, and ensure no head movement. Next, we use the HPI coils to localize the subject's head relative to the sensors. After head localization, the experimental portion of the run began. The experimental portion of the run consisted of two blocks each of safety and threat in alternating order. Each block contained eight presentations of the startle probe. Probes were separated by a variable interprobe interval (min = 6 s; max = 14 s), which was randomly determined for each probe. Shocks were presented during the threat blocks in randomly selected interprobe intervals, 2–4 s after the preceding probe. The number of shocks was randomly determined as well. Each run could contain between 0 and 2 shocks, and each interprobe interval had a 1 in 12 chance of containing a shock (unless the two shock ceiling had been reached).

### MRI

As with the MEG experiment, the subject arrived at the NIH clinical center, and completed the informed consent form, and pre-experiment questionnaire packet in the waiting room. Next subjects were escorted to the MRI control room and given the instructions for the task. Afterward, the subject was prepped for scanning. Electrodes to deliver the shock and record skin conductance were attached and secured to the subject. In addition, vitamin E capsules were attached to the subject in the same locations as the HPI coils during the MEG study, to facilitate coregistration of the structural MRI and MEG. Next the subject was escorted into the scan room, and given the shock workup procedure (*Balderston et al., 2017a*, *2017b*). Subjects were then given ear plugs, situated on the table, and connected to the pulse oximeter and breathing belt. Scanning began with a localizer, a T1-weighted structural scan, and two 30 s echo planar imaging (EPI) runs. These short EPI runs had opposing phase-encoding directions, and were used for EPI distortion correction (See below). After these, the subject was asked the pre-experiment affective rating scales over the intercom. Once complete, the subject began the experiment.

As in the MEG study, the subject viewed two concentric circles. The color of the outer circle indicated the type of block (orange = threat; blue = safe). The color of the inner circle indicated the subject's current level of anxiety. The instructions for the task were the same as the MEG study as well, with the exception of the difference in response device.

The experiment consisted of four runs, each of which were ~8 min (490 s) long. Each run consisted of two blocks each of safety and threat in alternating order. The number of shocks was randomly

determined as well. Each run could contain between 0 and 3 shocks at random intervals. There were no startle probes presented.

## Affective rating scales

Prior to each experiment, subjects completed several standard psychological questionnaires, including the Spielberger State-Trait Anxiety Index (STAI) (*Spielberger, 1987*), the Anxiety Sensitivity Index (ASI) (*Peterson and Heilbronner, 1987*), the Beck Anxiety Inventory (BAI) (*Beck et al., 1988*), and the Beck Depression Inventory (BDI) (*Beck and Steer, 1987*). At the start of both experiments and after each magnetoencephalography (MEG)/functional magnetic resonance imaging (fMRI) run, subjects were given a set of affective rating scales: (1) How anxious are you (1 = not anxious, 9 = extremely anxious)? (2) How afraid are you (1 = not afraid, 9 = extremely afraid)? (3) How would you rate the intensity of the electrical stimulation (1 = not painful at all, 9 = uncomfortable but not painful)?

## MRI acquisition

We collected four runs, each containing 245 multi-echo EPI images, using a 3T Siemens MAGNE-TOM Skyra (Erlangen,Germany) fMRI system, and a 32-channel head coil. For each image, we collected 32 interleaved 3 mm slices (matrix = 64 mm×64 mm; FOV = 192 × 192) parallel to the AC-PC line (TR = 2 s; TEs = 12 ms, 24.48 ms, 36.96 ms; flip angle = 70°). These 32 slices covered the entire cerebrum, but did not cover the most posterior parts of the brainstem and cerebellum. Slices were collected with an anterior-to-posterior phase encoding direction. We also collected two, 10 image multi-echo EPI series with the same parameters and the same field of view; however, one of these series was collected with a posterior-to-anterior phase encoding direction, and these series were used to correct for EPI distortion in the phase encoding direction (*Morgan et al., 2004*). We also acquired a multi-echo T1-weighted MPRAGE (TR = 2530 ms; TEs = 1.69 ms, 3.55 ms, 5.41 ms, 7.27 ms; flip angle = 7°). We acquired 176, 1 mm axial slices (matrix = 256 mm × 256 mm; field of view (FOV) = 256 mm × 256 mm), which were later co-registered to the EPI images.

## MRI preprocessing

Functional images were preprocessed and analyzed using the AFNI software package (see *Source code 1* for processing and analysis scripts) (*Cox, 1996*). EPI images for each run and each echo were first reconstructed, despiked (i.e. single voxel outliers were truncated), slice-time corrected, and then deobliqued. Then each volume in the series was registered to the first volume, and skull-stripped.

Preprocessed images were then entered into a multi-echo-independent components analysis using the meica.py script distributed with the AFNI software package (*Kundu et al., 2012*). This analysis uses the T2* decay of BOLD signals, measured across the echoes to denoise the timeseries. The analysis first decomposes the timeseries into independent components using FastICA. Then it determines whether signal intensity across echoes decays in a manner consistent with what is expected from BOLD data. Components that fit the model are kept, components that do not (i.e. components where the signal intensity does not decay across echoes) are discarded. A new denoised timeseries is then synthesized from the components not discarded. This technique has been previously shown to robustly remove sources of noise corresponding to motion, physiology, and scanner artifact (*Kundu et al., 2012*).

The denoised timeseries for all runs are then registered to the first run, scaled, and further denoised using a general linear model with regressors of no interest. Regressors of no interest included the six motion parameters from the volume registration step, up-through third-order polynomials to model baseline drift, and hemodynamic response functions (HRF)s corresponding to button presses and shock deliveries. In addition, images where the derivative of the motion regressors from volume registration step had a Euclidean norm above 0.5 mm were censored ('scrubbed') from further analyses. All remaining images from the safe and threat blocks were concatonated into separate timeseries; however, as part of the denoising procedure, we removed neural responses related to both button presses and transitions from one block type to another.

To correct for geometric distortion of the EPI images, the forward and reverse phase-encoding blips are first averaged across time, skull-stripped, and then rigid-body aligned with the reference

image for the other EPI timeseries. Then, the two blips are non-linearly aligned to each other using the 'plusminus' flag in the AFNI program 3dQwarp, so that the resulting image is 'in the middle' of the two. This reference image can then be registered to the T1 image, and the voxelwise displacement map for the forward blip is then saved, so that it can be applied to the EPI timeseries. Although a standard linear registration approach would have possibly yielded similar results at the group level, the nonlinear approach we used leads to better T1/EPI within-subject registration, and has been shown to perform even better than when using fieldmaps (*Hong et al., 2015*).

To align the EPI data to Montreal Neurological Institute (MNI) space and to mask out non-grey matter voxels, the T1 data was processed as follows. First, the volumes for the T1 echoes were averaged, and the resulting volume was run through the standard Freesurfer processing pipeline (*Desikan et al., 2006*; *Fischl et al., 2004*). Next, the skull-stripped anatomy is non-linearly registered to MNI space using the MNI_avg152T1 template distributed with AFNI. In addition, masks that included all cortical and sub-cortical grey matter atlas regions for each subject. These were warped to MNI space, downsampled to the EPI resolution, and dilated by 1 voxel. A group grey matter mask was created by averaging these binary masks, and thresholding out voxels with less than 2/3 overlap (*Torrisi et al., 2015*). Next the original space, skull-stripped anatomy was aligned to the distortion corrected EPI reference image in two steps. The first was a simple affine transformation. The second was a non-linear transformation using a criterion based on the local Pearson correlation (*Saad et al., 2009*), and the inverse displacement map was saved. Finally, the following warps were applied to the EPI timeseries data: inverse warp from the distortion correction step, affine transformation matrix from the T1 alignment step, the inverse warp from the T1 alignment step, and the non-linear warp from the T1 to MNI transformation. Finally, the EPI data were masked with the group grey matter mask, and blurred within this mask using a 6-mm FWHM Gaussian kernel.

## MRI analysis

To analyze the MRI recordings, we used a global brain correlation (GBC) approach (*Cole et al., 2010*). In brief, we correlated each voxel in our grey matter mask with each other voxel in the mask, applied the Fisher's Z transform, and summed across correlations. The result was a map where the value in each voxel reflected the strength of the correlation between that voxel and the rest of the brain. We conducted this GBC analysis for each subject and each condition, and used these values for further analysis. First, to identify changes in GBC across the entire brain, we averaged across voxels for safe and threat. We then conducted a paired-sample t-test on these averages. Next, to identify changes in GBC at the regional level, we analyzed the voxel-wise GBC for safe and threat. For this, we conducted a voxel-wise paired-sample t-test on the GBC values. Finally, to identify changes in connectivity with highly connected regions, we identified regions of interest (ROI)s from the voxel-wise GBC threat > safe t-test, and correlated the timecourse of activity in these ROIs with every voxel in the brain for safe and threat, and applied the Fisher's Z. We then conducted paired-sample t-tests on the resulting connectivity maps.

We used Monte Carlo simulations and a cluster-based method to correct for multiple comparisons across voxels. First, we estimated the smoothness in our residual timeseries using a Gaussian plus mono-exponential shaped function implemented by the '-acf' option in the AFNI program 3dFWHMx, which addresses recent concerns over inflated Type one error in studies using the cluster correction method (*Cox et al., 2016*). We calculated smoothness for each subject, and averaged this across subjects. Next, we simulated 10,000 random statistical parametric maps in 3dClustSim with a smoothness matching that of the original timeseries. For each simulation, we thresholded at a voxel-wise alpha of 0.005, and extracted the largest cluster. We then compared our test statistics to the distribution of clusters across all simulations to identify a minimum cluster size threshold of 80, corresponding to a two-tailed alpha of 0.05.

## MEG acquisition

We recorded neuromagnetic activity at 600 Hz from 271 radial first-order gradiometers using a 275 channel CTF-OMEGA whole-head magnetometer (VSM MedTech, Ltd., Canada). Recording took place in a magnetically shielded room (Vacuumschmelze, Germany), and Synthetic third-order gradient balancing was used for active noise cancellation (*Vrba and Robinson, 2001*).

## MEG preprocessing

MEG recordings were preprocessed and analyzed using the Fieldtrip toolbox in MATLAB (See *Source code 1* for processing and analysis scripts) (*Oostenveld et al., 2011*). First, movement within runs was checked by comparing the position of the HPI coils from the beginning of the run to the end of the run, and any run with a root mean square movement value above 1 cm was excluded from the analysis. Next, startle probe onsets were identified, and the 2 s window prior to each trial was extracted, demeaned, and detrended. Next, muscle artifacts were identified using the ft_artifact_muscle function in Fieldtrip. Trials where muscle movements were identified were removed. Next the recordings were low-pass filtered with a 90 Hz cutoff, and notch filtered at 60 Hz to remove line noise. The recordings were then downsampled to 300 Hz, and submitted to an independent components analysis. Components were visually inspected, and those with a topography, and time-course consistent with either blinks or heartbeats were identified for removal (typically 1–2 per artifact). Rejected components were projected out of the dataset, and the singular value decomposition of the covariance matrix was inspected to determine the regularization factor (lambda). A lambda of 5% was found to be sufficient to regularize the covariance matrix for source analysis following the removal of the rejected components. The result was a dataset cleaned for blink and heartbeat artifacts.

## MEG frequency analysis

Because we were specifically interested in alpha oscillations, we identified each subject's individual alpha frequency (IAF). We began by transforming the timeseries data into the frequency domain using a multi-taper fast Fourier transform (mtmfft) based on a set of discrete slepian sequences. In this initial transformation, we used a frequency window of 1 Hz to 20 Hz, and a single taper per frequency. We then averaged these spectrograms across sensors and across trials for each subject, and identified the largest local maxima in the average spectrogram for each subjects. For the majority of subjects (24/28), the largest peak occurred in the alpha frequency band (8–12 Hz). For all other subjects, the average IAF was used for further analyses. Next we conducted a second mtmfft, using a 2 Hz window centered around each subject's IAF, with two tapers per frequency and averaged the resulting power estimates across frequency for each sensor and each trial. This mean IAF power estimate was used in both the sensor space analyses and the source space analysis.

## MEG forward model

Single subject T1 images were used to generate the forward model for the MEG source analysis. First the T1 images were aligned to a single subject template in MNI space. Next the brain surface was extracted, and a single shell head model was generated from this surface. Then a source model was created using a single subject template with current dipoles placed along a regular 8 mm grid inside the brain surface. The single subject MNI space images were aligned to CTF space (i.e. coregistered to the sensors) manually, guided by the vitamin E capsules placed over the fiducial points. The resulting transformation matrix was applied to both the head model and the source model, and alignment between the sensors, head model, and source model was visually inspected. Finally, leadfields were then created using the location of the sensors, head model, and source model dipole locations.

## MEG inverse model

Because we were interested in frequency information (as opposed to time), we used the dynamic imaging of coherent sources (DICS) (*Gross et al., 2001*) technique to localize the sources of our recordings. We began by computing the cross spectral density matrix (CSD) from the frequencies of interest from all trials in the analysis. We then estimated the beamformer filter using the CSD, leadfields (with fixed orientations), headmodel, and gradiometer locations. Once estimated, this common filter was applied to the safe and threat conditions independently.

## MEG sensor-level analyses

We compared IAF power in safe vs. threat. For this, we averaged the IAF power across trials independently for safe and threat for each subject and each sensor, and conducted a paired sample t-test on these averages. For both analyses, we used Monte Carlo simulations and a cluster-based

method to correct for multiple comparisons across sensors. We calculated 1000 random permutations, where condition labels were shuffled across subjects. For each permutation, we thresholded the shuffled results at a sensor-level alpha of 0.005, summed the t-value across sensors in each cluster, and extracted the largest summed t-value. We then compared our test-statistic to the distribution of summed t-values and discarded any clusters where the summed t-value was smaller than the summed t-value corresponding to a two-tailed alpha of 0.05.

## MEG source-level analyses

As with the sensor-level analyses, to compare power in the safe and threat conditions, we projected the average IAF power into source space independently for each condition using the common filter calculated from all trials. We then conducted a paired-sample t-test on these power estimates at each voxel within the source model. As with the sensor-space data we used Monte Carlo simulations and a cluster-based method to correct for multiple comparisons across voxels. As before we calculated 1000 random permutations, where condition labels were shuffled across subjects. For each permutation, we thresholded the shuffled results at a source-level alpha of 0.005, summed the t-value across sensors in each cluster, and extracted the largest summed t-value. We then compared our test-statistic to the distribution of summed t-values and discarded any clusters where the summed t-value was smaller than the summed t-value corresponding to a two-tailed alpha of 0.05.

## Acknowledgements

The authors report no conflicts of interest. We would like to thank the NIMH Section on Instrumentation for fabricating the custom acoustic startle stimulus devices. Financial support of this study was provided by the Intramural Research Program of the National Institute of Mental Health, ZIAMH002798 (ClinicalTrial.gov Identifier: NCT00047853: Protocol ID 02 M-0321).

## Additional information

### Funding

| Funder | Grant reference number | Author |
| --- | --- | --- |
| National Institute of Mental Health | ZIAMH002798 | Christian Grillon |

The funders had no role in study design, data collection and interpretation, or the decision to submit the work for publication.

### Author contributions

NLB, Conceptualization, Software, Formal analysis, Investigation, Visualization, Methodology, Writing—original draft, Writing—review and editing; EH, AH, Investigation, Methodology, Writing—review and editing; ST, Conceptualization, Writing—review and editing; TH, FWC, Conceptualization, Software, Methodology, Writing—review and editing; RC, Conceptualization, Resources, Software, Methodology, Writing—review and editing; ME, Conceptualization, Supervision, Methodology, Writing—review and editing; CG, Conceptualization, Supervision, Investigation, Methodology, Writing—review and editing

### Author ORCIDs

Nicholas L Balderston, http://orcid.org/0000-0002-8565-1544

### Ethics

Human subjects: All participants gave written informed consent approved by the National Institute of Mental Health (NIMH) Combined Neuroscience Institutional Review Board and received compensation for participating.

## Additional files

**Supplementary files**
• Source code 1. Code used to conduct Startle, MEG, and MRI analyses. This zip file contains shell scripts and matlab functions that were used to analyze the Startle, MEG, and MRI data.

• Supplementary file 1. High resolution adjacency matrices for the MEG connectivity analyses. This zip file contains high resolution images of the adjacency matrices for the MEG connectivity analysis suggested by the editor and reviewers.

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
