## [Decision Letter]

Thank you for submitting your article "Threat of shock increases excitability and connectivity of the intraparietal sulcus" for consideration by *eLife*. Your article has been reviewed by three peer reviewers, one of whom is a member of our Board of Reviewing Editors, and the evaluation has been overseen by David Van Essen as the Senior Editor. The following individual involved in review of your submission has agreed to reveal their identity: Krish Singh (Reviewer #3).

The reviewers have discussed the reviews with one another and the Reviewing Editor has drafted this decision to help you prepare a revised submission.

Summary:

Your paper was assessed positively by all three reviewers, with all three saying that this is a valuable addition to the literature and of broad scientific interest. In particular, the multi-modal aspect was well received. I am therefore happy to provisionally recommend publication, however a number of significant revisions must be successfully carried out first. On receiving your revised manuscript with explanatory comments, the Reviewing Editor will decide whether the manuscript needs to be seen again by the reviewers.

Essential revisions:

For fMRI, all of the technical concerns raised by reviewer 2 must be addressed comprehensively. These are copied in full below.

For MEG, both reviewers 1 and 3 suggest that a functional connectivity analysis would be a better candidate for comparison with your fMRI results than the current approach. I therefore strongly recommend that you undertake such analyses; here, a key confound is signal leakage between source space regions and an appropriate correction method must be employed. Though multiple schemes are available I would suggest using the approach published by Colclough, Brookes, Smith and Woolrich, 2015. (referenced by reviewer 1) but perhaps applied to a denser parcellation (e.g. the AAL parcellation). Again for completeness the reviewers comments are copied in full below.

Finally, a quantitative analysis of the similarity of the regions identified by MEG and fMRI must be given.

Reviewer #1:

The paper by Balderston et al. describes a study in which both fMRI and MEG are used to investigate the neural correlates of anxiety. To this end, a threat of shock paradigm is employed to increase anxiety in subjects; this increase is then assessed behaviourally in multiple ways. Both fMRI and MEG data are acquired during the paradigm and analysed in an unbiased way; briefly, in fMRI an 'all-to-all' connectivity approach is used to measure connectivity among voxels. In MEG, data are analysed in multiple frequency bands and changes in oscillatory power between task blocks (safe versus threat of shock) characterised. The interesting result is that both the fMRI connectivity analysis and the α band power assessment both implicate the intraparietal sulcus as a key hub in processing anxiety.

Overall I believe this to be a very interesting and exciting paper. The primary finding is not only impactful for basic science but may also, potentially, have significant clinical relevance. The paper is well written and the analyses, for the most part, solid. However I do have a number of suggestions that I think would improve the paper.

Major comments

1) It is assumed that the fMRI and MEG responses occur in the same location, however this was never analysed quantitatively. The distance between the fMRI cluster and the MEG peak location should be quantified. What are the chances that this difference could occur by chance?

2) It is not made clear in the Introduction why complementary analysis is not used – e.g. the MEG is used to assess oscillatory changes whereas fMRI is used to assess connectivity. Why? I realise of course the links that have been made between oscillations and connectivity but this is not well discussed in the paper (I personally didn't find the relation to the simultaneous EEG-fMRI literature very convincing). Why not just do a standard GLM approach to the BOLD analysis to get regions of increased BOLD response during threat and compare that with changes in α oscillations? In short, the analysis pipeline used should be better motivated.

3) Related to the above question, I found the MEG analysis quite limited. Wouldn't it have been better to take advantage of the huge steps forward in MEG connectivity methods that have been made recently and employ one such approach to measure whole brain α band connectivity – and then compare this to the fMRI connectivity. Its true that you couldn't do this at the voxel level spatial scale but with a brain parcellation it should be easily possible. See papers by e.g. Colclough, Brookes, Smith and Woolrich, 2015 or O'Neill et al., http://iopscience.iop.org/article/10.1088/0031-9155/60/21/R271

4) I felt there should be more justification for looking at the α band, which seems to have been chosen specifically, with the other frequency bands treated as an afterthought. Given the apparent close links between β oscillations and functional connectivity why was the β band not chosen? Again just saying that α was the strongest signal at many sensors didn't really convince me.

5) Can the authors give some explanation as to why the α response was lateralised?

Reviewer #2:

General

This study is a nice example of an investigation into neuronal mechanisms of anxiety that uses two different complementary modalities. However, some of the fMRI analysis approaches need further investigation to ensure that the results were not driven by noise, and the manuscript is longer than necessary in some places.

Major comments

1) The within-network results do not pass multiple comparison correction and do not add much to the story. I would therefore suggest that the authors remove these results. In general, the manuscript should be shortened. Specifically the Discussion would benefit from being more concisely written, and the Materials and method could also be reduced (specifically the subject numbers and the 'on the day of the appointment…' text).

2) Was there a difference in motion parameters between the safe and threat blocks in fMRI? If so, this could lead to a shift between short-distance and long-distance connectivity, which could drive the results. The scrubbing that was performed most likely avoids this possibility, but it would be of interest to present the difference in motion parameters, and to potentially run further analyses to explicitly exclude this possibility (for example, by selecting periods or subjects in which the motion was matched between the two conditions).

3) How many time points were removed during scrubbing, and was there a significant difference between the safe and shock blocks for this? If so, the difference in power could bias the results, and it would be worth excluding this possibility by matching the number of timepoints included in shock and safe blocs (within participants).

4) The change in global connectivity during the shock period could occur as a result of the actual shock stimuli that were not present in the safe periods. I know the HRF responses to shocks and button presses were modeled and removed, but this may not capture the changes in functional connectivity resulting from shock stimuli. Would it be possible to repeat the analysis using only shock blocks that contained zero shocks to ensure that this is not driving the results?

5) Cluster-based corrections with an initial cluster forming threshold above 0.001 have been shown to suffer from inflated false positives (Eklund et al., 2016). The authors should at least point this out in the limitations, and should ideally repeat analyses using the latest guidelines.

6) The methods used in this study are described as unbiased and multimodal, which is a strong statement to make given that some of the whole-brain statistics might suffer from a false positive bias, and the different modalities are analysed separately, rather than in a joint multimodal approach.

Reviewer #3:

I really enjoyed reading this paper, which is a valuable contribution to the field. It uses a multi-modal approach of combining fMRI and MEG to reveal changes in both connectivity across the brain (using fMRI) and oscillatory power changes (using MEG) while participants were either in a state of 'Safe' or 'Threat'

To me the results are interesting both in terms of probing the neurophysiological underpinnings of threat/anxiety but also the relationship between fMRI and MEG measures of brain function.

The methods are thorough and well-executed and reveal a good understanding of the state-of-the-art in fMRI and MEG analyses. However I think more could be done with the MEG data (see my comment 1 below).

The paper is well-written and readable and the authors nicely discuss the weaknesses of their approach.

Here are my main comments:

1) It seems as if there is a missed opportunity here in terms of directly comparing connectivity measures in fMRI to those extracted with MEG (albeit at the group-level). Using a sub-sampled atlas approach (say the AAL atlas) the authors could easily compare 'GBC"-type connectivity matrices extracted with fMRI with those extracted with MEG, either using amplitude-amplitude coupling within frequency bands or phase-phase coupling. There are several recent papers using this approach, so it seems odd that the authors did not do this.

2) It is not absolutely clear in the Abstract/Introduction that when the authors talk about neural activity and connectivity in the IPS, they are really talking about haemodynamics measures (i.e. FMRI). This should be made clearer.

3) When presenting the fMRI-GBC (for example in Figure 3) I think it would be useful to show the spatial distribution across the brain of the connectivity measure i.e. after summing across rows but before global summing across the brain. This would be a companion figure to Figure 4, which shows the voxelwise difference in connectivity strength between Threat v. Safe. Based on my comment 1 above it would be great to see similar visualisations based on frequency-specific MEG connectivity maps.

4) In the MEG analysis, it looks like some trials are rejected if contaminated with artefacts. I just wanted to check that the statistical tests performed later (between Safe and Threat) properly account for differing number of trials.

5) In the MEG analysis, did the authors check that there was no difference in head movement between the Safe and Threat blocks?

[Editors' note: further revisions were requested prior to acceptance, as described below.]

Thank you for resubmitting your work entitled "Threat of shock increases excitability and connectivity of the intraparietal sulcus" for further consideration at *eLife*. Your revised article has been favorably evaluated by David Van Essen (Senior editor), a Reviewing editor, and two reviewers.

The manuscript has been improved but there are some remaining issues that need to be addressed before acceptance, as outlined below:

We thank the authors for their careful consideration of the comments of all three reviewers, and for the extra analyses that have been undertaken. The paper is much improved as a result of these changes and is now close to publication. However, a few points remain to be addressed.

1) I consider the conjunction map, provided in the response to reviewers, to be critical to the paper and I would very much like to see this in the main paper, perhaps added to Figure 7.

2) Its rather a shame that the MEG connectivity analysis didn't work, but nevertheless I thank the authors for attempting it. I wonder however if some extra clarification could be given: What frequency band was connectivity computed in (sincere apologies if I missed it)? I assume α? I'm also unsure as to what is meant by "we then computed the Hilbert transform using a 500 ms sliding window"? The Hilbert transform shouldn't need a sliding window – when undertaking this analysis we usually compute the HT over all time? Please clarify why this was done. Could the authors please show the MEG adjacency matrices for the safe and thread blocks independently, as well as the difference? This would allow the reader to confirm that the two separate matrices look sensible (one expects large occipital connectivity in the α band, see e.g. Hunt et al., 2016). Finally, given the strong finding of MEG connectivity in the β band, I would like to see the connectivity analysis attempted in this band. I of course read the authors argument that "one might expect a high degree of stationarity in coherence of the β oscillations across time" but I disagree, specifically because multiple papers (e.g. O'Neill et al., 2015; O'Neill et al., 2016; Baker et al., 2014) show that, in fact, β band functional connectivity is highly dynamic.

[Editors' note: further revisions were requested prior to acceptance, as described below.]

Thank you for resubmitting your work entitled "Threat of shock increases excitability and connectivity of the intraparietal sulcus" for further consideration at *eLife*. Your revised article has been favorably evaluated by David Van Essen (Senior editor) and the Reviewing editor.

The manuscript has been improved but there are some remaining issues that need to be addressed before acceptance, as outlined below:

I thank the authors for going to such lengths on the MEG connectivity analysis. However, something has clearly gone wrong here in the analysis.

By my assessment of the adjacency matrices presented, the case for the α and β bands look virtually identical. However this should not be the case (again see Hunt et al, 2016). It is quite hard for me to judge the spatial signature of connectivity as, unfortunately, in the pdf version the figure quality is too low resolution to read the region names on the axes. However, these matrices do not look correct to me and I would urge the authors to find out why.

In their response, the authors have suggested that they use blocks of only 2 seconds of envelope data. However they also apply downsampling prior to connectivity estimation. This to me seems silly – why apply the downsampling if you have such short blocks. It is well established that connectivity works without the downsampling so perhaps removing that step might improve the adjacency matrices. I also don't understand why such short data setments were used when in fact the blocks are quite long. Why not just try using the whole block?

I remain enthusiastic about this article, but I maintain that the MEG connectivity analysis has not yet been properly carried out. I would want to see this estimated reliably (or a concrete argument on why it cannot be undertaken) put forward before recommending publication.

[Editors' note: further revisions were requested prior to acceptance, as described below.]

Thank you for resubmitting your work entitled "Threat of shock increases excitability and connectivity of the intraparietal sulcus" for further consideration at *eLife*. Your revised article has been favorably evaluated by David Van Essen (Senior editor) and the Reviewing editor.

The manuscript has been improved but there are some remaining issues that need to be addressed before acceptance, as outlined below:

I would like to thank the authors for their attempt at addressing my concerns. I'm pleased that the authors were able to spot errors in their pipeline (use of incorrect spatial filters) although it does appear that the major difference came from the down-sampling, as suggested in my previous review. Downsampling over such a small time window is obviously incorrect so these analyses should now be discarded.

Unfortunately, the adjacency matrices still don't look right – without seeing them plotted on a brain it's hard to judge exactly where the primary pathways of connectivity are (and I still can’t read the labels clearly in the pdf). As I have said previously, the highest connectivity in the α band should be in the occipital lobe whilst the highest connectivity in the β band should encompass bilateral parietal and occipital connections alongside tempero-parietal and fronto-parietal networks (again as in Hunt et al, 2016). In fact this does not seem to be the case. In what the authors have provided the α network looks like pure noise. There is clearly some structure to the β band connectivity matrix although this doesn't really look like one would expect. So I strongly suspect something is still wrong with the analysis.

Without looking at the data directly it’s hard for me to judge but I suspect that the problem is the short time windows (unless there are other more basic errors (similar to the incorrect spatial filter) which the authors have not discovered). It is known that a reasonable amount of data is required to make MEG connectivity analyses work reliably (Luizzi et al., Optimising experimental design for MEG resting state functional connectivity measurement, NeuroImage 2016) and perhaps this is the reason the adjacency matrices look so poor. In the light of these failed attempts, I suggest that a paragraph is added to the Discussion stating that MEG connectivity analysis would have been a useful means to probe these data; however the study design was poorly set up to make such analyses work.

Please note that this must now be your final attempt to satisfy the Board.

---

## [Author Response]

*Essential revisions:*

*For fMRI, all of the technical concerns raised by reviewer 2 must be addressed comprehensively. These are copied in full below.*

Reviewer 2 raised 6 technical concerns in their major issues section. Technical concern 1 was addressed as part of a larger effort to shorten the manuscript, and therefore will be described below in a separate section about manuscript length. Technical concerns 2-4 were all related to ruling out alternative explanations for our global connectivity results. These will be addressed together in the paragraphs below. Finally, technical concerns 5 and 6 were separate issues, and are addressed inline with the reviewer’s original comments.

In technical concerns 2-4, reviewer 2 questioned whether the following could explain our global connectivity results: 2) the difference in motion across block types (i.e. safe vs. threat), 3) the difference in the number of TRs censored across block types, and 4) the presentations of the shocks in the threat blocks. Before we describe the additional steps conducted to rule out these possibilities, we would like to note that we took very thorough steps in the original analysis to account for possible sources of noise in the fMRI data. First, we used the novel multi-echo ICA cleaning approach which identifies and eliminates sources of noise that do not exhibit the expected TE-dependence of the BOLD signal. Second, we used a strict threshold for head motion scrubbing (i.e. censoring TRs with a framewise displacement > 0.5 mm). Third, we used a GLM to remove noise related to the 6 motion parameters from the EPI timeseries. Fourth, we modeled the HRF corresponding to each shock delivery, and included this as a regressor of no interest in above mentioned GLM.

To further address these questions, we began by examining whether motion and TRs censored differed across block type in the original analysis. To quantify motion, we calculated the Euclidean norm of framewise displacement across the six motion parameters after scrubbing, and averaged this across the TRs included in the analysis for each block type. We found that this indeed differed across the safe and threat blocks (t(24) = 15.32; p < 0.001). Similarly, when we count the total TRs included in the analysis (Total TRs – Censored TRs) we found that this also differed across threat and safe blocks (t(24) = -6.89; p < 0.001). Therefore to account for this difference, and to account for the potential effects of the shock delivery, we recalculated global brain connectivity (GBC) using a more selective set of TRs.

In this analysis, we censored the 10 TRs during the threat periods following each shock delivery, and then censored safe TRs so that the total number of TRs was equivalent across the safe and threat blocks, and the Euclidean norm of framewise displacement was as equivalent as possible across the safe and threat blocks (See Figure 9 for total number of TRs and framewise displacement before and after this correction). Although we were able to fully match the total number of TRs across blocks, the framewise displacement was not fully matched (t(24) = -3.75; p = 0.001). However, it should be noted that the average framewise displacement after this correction was very low (< 1mm). To address this remaining difference in framewise displacement, we calculated the difference in framewise displacement across the safe and threat blocks, and used this value as a covariate in an ANCOVA looking at the effect of threat on GBC across this more selective set of TRs.

Author response image 1.**DOI:**
http://dx.doi.org/10.7554/eLife.23608.022

Since the purpose of this analysis was to determine whether our original results remained after this more stringent scrubbing, we sampled GBC within the same ROIs identified by the original whole brain GBC analysis. We found that even after censoring out 10 TRs following the shock, matching the total number of TRs censored, closely matching framewise displacement, and factoring out the remaining differences in framewise displacement, we still see a significant difference in GBC as a function of threat in all 3 ROIs identified in the original analysis. These results suggest that our initial findings were not due to differences in motion, censoring, or residual neural activity evoked by the shock. We understand that these are important controls, and have therefore revised the original version of the manuscript to include the details of this analysis.

*For MEG, both reviewers 1 and 3 suggest that a functional connectivity analysis would be a better candidate for comparison with your fMRI results than the current approach. I therefore strongly recommend that you undertake such analyses; here, a key confound is signal leakage between source space regions and an appropriate correction method must be employed. Though multiple schemes are available I would suggest using the approach published by Colclough Brookes, Smith and Woolrich, 2015. (referenced by reviewer 1) but perhaps applied to a denser parcellation (e.g. the AAL parcellation). Again for completeness the reviewers comments are copied in full below.*

We agree with the reviewers that this work would benefit from a connectivity analysis of the MEG data, and recognize that signal leakage is a serious issue affecting such analysis. Therefore, we have chosen to follow the reviewers’ advice and conduct an analysis of the differences in MEG connectivity as a function of threat, while accounting for signal leakage using the symmetric orthogonalisation method detailed by Colclough, Brookes, Smith and Woolrich, 2015.

We began by computing the virtual MEG channels using the single trial timecourses and leadfields computed in the original manuscript. Then, using the toolbox provided by Dr. Colclough (https://github.com/OHBA-analysis/MEG-ROI-nets), we downsampled the source space MEG data to the AAL atlas, and computed the symmetric ortrhogonalization of the atlas space signals to remove any 0-phase correlations. We then computed the Hilbert transform using a 500 ms sliding window, low-pass filtered the data to 0.5 Hz, and downsampled the data to 1 Hz. Next we concatonated these envelope timeseries across all trials for safe and threat separately for each subject, and computed the cross correlation matrix for each subject and each block separately. To determine whether there were any differences in connectivity between safe and threat, we computed paired-sample t-tests at each cell of the cross correlation matrix. We then conducted 10,000 permutation tests and extracted the maximum t-value for each permutation in order to build a null distribution for these t-tests. Finally, we identified a corrected t-value at the 95 percentile of this distribution (t = 5.5), and thresholded the original t-map using this corrected t-value. As can be seen from the graphs in Figure 10, there were no significant changes in pairwise functional connectivity as a function of threat based on this analysis and correction method.

Author response image 2.**DOI:**
http://dx.doi.org/10.7554/eLife.23608.023

Given that the overall suggestion was to reduce the length of the manuscript, we have decided not to include this analysis in the revised submission. We have however revised the limitations section of the manuscript to include a discussion of this point. We are happy to make additional changes if requested.

*Finally, a quantitative analysis of the similarity of the regions identified by MEG and fMRI must be given.*

We agree with this reviewer that it is important to demonstrate the correspondence between the regions identified by the MEG and fMRI analyses. Accordingly, we have prepared a conjunction map of these two results, Figure 11. There are two major points illustrated by this map. First, although the fMRI results are bilateral, the MEG results are unilateral. This point will be addressed below. Second, the fMRI cluster in the left IPS substantially overlaps with the source space MEG cluster (46 of 81 voxels). Due to the recommendation to reduce the length of the manuscript, we did not include this figure in the revised version of the manuscript. However, we did revise the Results section to reflect this finding. If requested, we would be happy to add a figure with this result to the manuscript.

Author response image 3.**DOI:**
http://dx.doi.org/10.7554/eLife.23608.024

*Reviewer #1:*

[…]

*Major comments*

*1) It is assumed that the fMRI and MEG responses occur in the same location, however this was never analysed quantitatively. The distance between the fMRI cluster and the MEG peak location should be quantified. What are the chances that this difference could occur by chance?*

This point has been addressed in the general comments to the editor, see above.

*2) It is not made clear in the Introduction why complementary analysis is not used – e.g. the MEG is used to assess oscillatory changes whereas fMRI is used to assess connectivity. Why? I realise of course the links that have been made between oscillations and connectivity but this is not well discussed in the paper (I personally didn't find the relation to the simultaneous EEG-fMRI literature very convincing). Why not just do a standard GLM approach to the BOLD analysis to get regions of increased BOLD response during threat and compare that with changes in α oscillations? In short, the analysis pipeline used should be better motivated.*

We agree with this reviewer that a standard GLM approach to analyze the BOLD data might be a better match for the analysis of the MEG oscillation data. However, our blocks were quite long for fMRI data ~2 min, and therefore subject to baseline drifts. Although there are approaches to minimize the contributions of these baseline drifts to activation estimates (including the multi-echo ICA denoising technique used here), it is unclear whether these techniques are robust enough to warrant conducting the additional GLM approach suggested. Given the recommendation to reduce the overall length of the manuscript, we have decided not to include this analysis. However, we are happy to conduct the GLM analysis, and include it in the paper if deemed necessary.

3) Related to the above question, I found the MEG analysis quite limited. Wouldn't it have been better to take advantage of the huge steps forward in MEG connectivity methods that have been made recently and employ one such approach to measure whole brain α band connectivity – and then compare this to the fMRI connectivity. Its true that you couldn't do this at the voxel level spatial scale but with a brain parcellation it should be easily possible. See papers by e.g. Colclough, Brookes, Smith and Woolrich, 2015 or O'Neill et al., http://iopscience.iop.org/article/10.1088/0031-9155/60/21/R271

We agree that MEG connectivity is an interesting topic, and have undertaken the analysis described in Colclough, Brookes, Smith and Woolrich, 2015. This analysis is described in the general comments to the editor, see above.

*4) I felt there should be more justification for looking at the α band, which seems to have been chosen specifically, with the other frequency bands treated as an afterthought. Given the apparent close links between β oscillations and functional connectivity why was the β band not chosen? Again just saying that α was the strongest signal at many sensors didn't really convince me.*

We agree that there is a strong connection between functional connectivity and β oscillations. Indeed, a careful examination of spontaneous β oscillations can uncover a set of resting state networks that is remarkably similar to those identified from spontaneous fluctuations in BOLD data. However, given that resting state networks identified from BOLD data are highly correlated with independent measures of anatomical connectivity, one might expect a high degree of stationarity in coherence of the β oscillations across time. This might be acceptable if the purpose of this study were to identify correlations between resting state networks and anxiety across subjects. However, the purpose of this study was to determine how a within-subject experimental manipulation of anxiety affects spontaneous measures of neural activity/connectivity.

Although the focus of the paper was on α, our initial approach (described in the Introduction) was to examine all frequency bands independently. The focus on α emerged out of the observations that, 1) α was the strongest signal in the recordings, 2) α showed the largest power changes as a function of threat, 3) α was the only frequency band that showed consistent results at both the sensor and the source level, 4) the source space results aligned nicely with the corresponding fMRI GBC data.

It should also be noted that there are several theoretical reasons to focus on α as well. For instance, it is well established that spontaneous neural activity at rest is dominated by oscillations in the α (8-12 Hz) range, which are most prominent when the subject is in an alert state of restful relaxation, and that α asymmetries can reflect differences in arousal across subjects. Theoretical models of α function suggest that α oscillations are generated by coherent activity in local inhibitory interneurons, and that decreases in α power reflects increases in cortical excitability. Consistent with these theories, studies collecting simultaneous measures of electroencephalography (EEG) and fMRI have shown that α power is negatively correlated with functional connectivity.

Given these points, we believe that it is appropriate to focus primarily on the α results. However, we agree that this motivation should be clarified. This is especially true given that we have chosen to remove the results from the other frequency bands, in order to respond to the request to shorten the manuscript. Therefore we have revised the Discussion section of the manuscript to clarify our motivation for focusing primarily on the α results.

*5) Can the authors give some explanation as to why the α response was lateralised?*

There is actually a robust literature on α asymmetry and emotional processing. According to prominent models, left dominant frontal α is associated with positive affect and/or approach behavior, while right dominant α is associated with negative affect and/or avoidance behavior (Davidson 2004; Harmon-Jones, Gable and Peterson, 2010). Like the valence model of α asymmetry, one prominent model on parietal α asymmetry is rooted in the arousal-valence model of emotional processing (Heller, 1993). According to this model, right dominant parietal α is associated with increased arousal. Consistent with this theory, we find that threat of shock, which increases arousal, also reduces left parietal α, resulting in a right dominant profile. However, more research should be conducted to specifically test this hypothesis. We understand that this is an important point, and have revised the manuscript to include a brief discussion of this aspect of the findings.

*Reviewer #2:*

*General*

*This study is a nice example of an investigation into neuronal mechanisms of anxiety that uses two different complementary modalities. However, some of the fMRI analysis approaches need further investigation to ensure that the results were not driven by noise, and the manuscript is longer than necessary in some places.*

*Major comments*

*1) The within-network results do not pass multiple comparison correction and do not add much to the story. I would therefore suggest that the authors remove these results. In general, the manuscript should be shortened. Specifically the Discussion would benefit from being more concisely written, and the Materials and method could also be reduced (specifically the subject numbers and the 'on the day of the appointment…' text).*

This point has been addressed in the general comments to the editor, see above.

*2) Was there a difference in motion parameters between the safe and threat blocks in fMRI? If so, this could lead to a shift between short-distance and long-distance connectivity, which could drive the results. The scrubbing that was performed most likely avoids this possibility, but it would be of interest to present the difference in motion parameters, and to potentially run further analyses to explicitly exclude this possibility (for example, by selecting periods or subjects in which the motion was matched between the two conditions).*

As with point 1, this point has been addressed in the general comments to the editor, see above.

*3) How many time points were removed during scrubbing, and was there a significant difference between the safe and shock blocks for this? If so, the difference in power could bias the results, and it would be worth excluding this possibility by matching the number of timepoints included in shock and safe blocs (within participants).*

As with point 1 and 2, this point has been addressed in the general comments to the editor, see above.

*4) The change in global connectivity during the shock period could occur as a result of the actual shock stimuli that were not present in the safe periods. I know the HRF responses to shocks and button presses were modeled and removed, but this may not capture the changes in functional connectivity resulting from shock stimuli. Would it be possible to repeat the analysis using only shock blocks that contained zero shocks to ensure that this is not driving the results?*

As with points 1, 2, and 3, this point has been addressed in the general comments to the editor, see above.

*5) Cluster-based corrections with an initial cluster forming threshold above 0.001 have been shown to suffer from inflated false positives (Eklund et al., 2016). The authors should at least point this out in the limitations, and should ideally repeat analyses using the latest guidelines.*

Indeed, we agree that several common approaches to multiple comparisons suffer from inflated Type 1 error, as indicated by Eklund et al., 2016. Cluster thresholding, as implemented by the AFNI program 3dClustSim, had, until recently, used Gaussian estimates of the noise smoothness in the monte carlo simulations, which do not appropriately model the noise smoothness of BOLD data. The AFNI team has since provided new option that models the noise smoothness using a Gaussian plus mono-exponential spatial AutoCorrelation Function (ACF), which addresses the criticisms raised by the Eklund paper.

See:

http://biorxiv.org/content/early/2016/07/26/065862

In the current work, we used the newly created autocorrelation function to estimate the smoothness of the noise in our data, which was used to identify minimum cluster-size threshold reported in the manuscript. We realize that this was not sufficiently explained in the original version of the manuscript, and have therefore clarified this methodological detail in the revised version.

In addition, although our voxelwise p-value is higher than recommended by the reviewer, this is offset by a larger minimum cluster-size threshold. This, combined with the fact that our results are bilateral and consistent with the MEG data, leads us to conclude that it is unlikely that our results are due to Type 1 error.

*6) The methods used in this study are described as unbiased and multimodal, which is a strong statement to make given that some of the whole-brain statistics might suffer from a false positive bias, and the different modalities are analysed separately, rather than in a joint multimodal approach.*

The main finding of the current work is that aspects of IPS functioning are affected by the threat of shock manipulation. We show this in two ways: first we show that α oscillations are decreased by threat of shock (MEG), second we show that global IPS functional connectivity is increased by threat of shock (fMRI). These two independent findings are significant at the whole brain level, corrected for multiple comparisons. For that reason, we feel that it is appropriate to refer to these findings as both unbiased and multimodal.

*Reviewer #3:*

[…]

*Here are my main comments:*

*1) It seems as if there is a missed opportunity here in terms of directly comparing connectivity measures in fMRI to those extracted with MEG (albeit at the group-level). Using a sub-sampled atlas approach (say the AAL atlas) the authors could easily compare 'GBC"-type connectivity matrices extracted with fMRI with those extracted with MEG, either using amplitude-amplitude coupling within frequency bands or phase-phase coupling. There are several recent papers using this approach, so it seems odd that the authors did not do this.*

This point has been addressed in the general comments to the editor, see above.

*2) It is not absolutely clear in the Abstract/Introduction that when the authors talk about neural activity and connectivity in the IPS, they are really talking about haemodynamics measures (i.e. FMRI). This should be made clearer.*

In the Introduction and Abstract we refer to activity and connectivity. When we use the term activity, we are referring to reductions in α power (MEG) that reflect increases in excitability. When we use the term connectivity, we are referring to the functional connectivity measured with fMRI. These points have been clarified in the revised version of the manuscript (See Abstract and Introduction).

*3) When presenting the fMRI-GBC (for example in Figure 3) I think it would be useful to show the spatial distribution across the brain of the connectivity measure i.e. after summing across rows but before global summing across the brain. This would be a companion figure to Figure 4, which shows the voxelwise difference in connectivity strength between Threat v. Safe. Based on my comment 1 above it would be great to see similar visualisations based on frequency-specific MEG connectivity maps.*

We agree that the revised version of the manuscript would benefit from this figure. Therefore, we have updated Figure 3 to reflect this change. As for the MEG connectivity maps, we conducted the functional connectivity analysis recommended by the editor, which yielded no significant results. Given the overall recommendation to shorten the manuscript, we have decide not to include these results in the final work. However, we are happy to add this work if deemed necessary.

*4) In the MEG analysis, it looks like some trials are rejected if contaminated with artefacts. I just wanted to check that the statistical tests performed later (between Safe and Threat) properly account for differing number of trials.*

Indeed, when we examine the number of trials included in the analysis across conditions, we do see a difference in the number of trials for the safe and threat conditions. In the safe condition there were on average 58.21 ± 5.5 trials, while in the threat condition there were on average 55.32 ± 7.6 trials, which was significantly different across subjects (t(124) = 2.78, p = 0.01). Therefore, we decided to determine whether this difference in trial number impacted our results at the censor and source level. Accordingly, we included the difference in trial number across safe and threat blocks as a covariate in an ANCOVA examining the effect of threat on α responses. At both the sensor (f(1,26) = 7.48, p = 0.01) and at the source (f(1,26) = 17.797, p < 0.001), we find that even with the difference in number of trials covaried out, the effects of threat on α power is still significant, suggesting that this difference did not significantly impact our results. This analysis has been included in the revised version of the manuscript.

5) In the MEG analysis, did the authors check that there was no difference in head movement between the Safe and Threat blocks?

Unfortunately we are unable to determine this from our current data. In this study we determined the amount of motion by comparing the subject’s position before a run to their position after the run. Accordingly, we did not have an online measure of motion within a given run. Each run contained four blocks: two safe and two threat, so we do not have condition specific measures of motion for the MEG study. We agree that this is a limitation of the current work, and have added a statement to the Discussion to address this limitation.

[Editors' note: further revisions were requested prior to acceptance, as described below.]

*We thank the authors for their careful consideration of the comments of all three reviewers, and for the extra analyses that have been undertaken. The paper is much improved as a result of these changes and is now close to publication. However, a few points remain to be addressed.*

*1) I consider the conjunction map, provided in the response to reviewers, to be critical to the paper and I would very much like to see this in the main paper, perhaps added to Figure 7.*

We agree that this conjunction map is an important part of the work, and are happy to add it to the main paper. We have added it as Figure 8 in the revised version of the manuscript.

*2) Its rather a shame that the MEG connectivity analysis didn't work, but nevertheless I thank the authors for attempting it. I wonder however if some extra clarification could be given: What frequency band was connectivity computed in (sincere apologies if I missed it)? I assume α? I'm also unsure as to what is meant by "we then computed the Hilbert transform using a 500 ms sliding window"? The Hilbert transform shouldn't need a sliding window – when undertaking this analysis we usually compute the HT over all time? Please clarify why this was done. Could the authors please show the MEG adjacency matrices for the safe and thread blocks independently, as well as the difference? This would allow the reader to confirm that the two separate matrices look sensible (one expects large occipital connectivity in the α band, see e.g. Hunt et al., 2016). Finally, given the strong finding of MEG connectivity in the β band, I would like to see the connectivity analysis attempted in this band. I of course read the authors argument that "one might expect a high degree of stationarity in coherence of the β oscillations across time" but I disagree, specifically because multiple papers (e.g. O'Neill et al., 2015; O'Neill et al. 2016; Baker et al. 2014) show that, in fact, β band functional connectivity is highly dynamic.*

There were several points raised in the above comment. For clarity, we have numbered these, and addressed them below.

a) This is correct. Given that the other analyses in the paper were focused on the α band, we conducted the connectivity analyses in the α band as well.

b) This statement was made in error. We were describing the functionality of the power envelope calculation implemented in the MEG-ROI-nets toolbox, and mistakenly mixed up the steps. Upon reviewing the methods of the Colclough, Brookes, Smith and Woolrich, 2015, and the matlab functions provided by Dr. Colclough on github (https://github.com/OHBA-analysis/MEG-ROI-nets), it is clear that these functions compute the power envelope by first taking the Hilbert transform, then low-pass filtering the data, and finally downsampling the data using a sliding window approach.

c) We have attached the adjacency matrices for the safe and threat conditions in Figure 12. Although the matrices do not exactly match those from Hunt et. al 2016, there appears to be α connectivity in the occipital lobe for both safe and threat conditions. One possible reason for the difference across studies is that we calculated connectivity across a series of discrete events (the 2 seconds prior to each startle probe). We understand that these short blocks are not optimal for studying connectivity; however, the study was initially designed to investigate pre-stimulus activity and startle magnitude under threat.

Author response image 4.Α connectivity across AAL regions during 2 second baseline prior to startle probe.**A**) Mean α connectivity across AAL regions during safe periods. **B**) Mean α connectivity across AAL regions during threat periods. **C**) Unthresholded T-test results comparing α connectivity during safe and threat conditions. Thresholded T-test results comparing α connectivity during safe and threat conditions. Labels on Y-axis correspond to regions of the AAL axis. Labels on the X-axis correspond to groups from AAL atlas (frontal, limbic, occipital, parietal, subcortical, temporal, cerebellum). Boxes in **A** and **B** represent α connectivity in the occipital cortices.**DOI:**
http://dx.doi.org/10.7554/eLife.23608.025

d) As requested, we have conducted an initial analysis of connectivity in the β band. We used the exact approach as in the previous study. Briefly, we bandpass filtered the data, downsampled it to the AAL atlas, applied the orthogonalization procedure detailed in Colclough, Brookes, Smith and Woolrich, 2015, computed the Hilbert transform, downsampled the power envelopes, and computed pairwise t-tests at each node of the adjacency matrices. Note that, like α, there were no significant differences in β connectivity from safe to threat after correcting for multiple comparisons. Although it may be interesting to explore changes in MEG connectivity as a function of threat, perhaps future studies can be conducted to examine this question. See Figure 13.

Because the connectivity analysis involves downsampling the power envelopes, one recommendation for future studies would be to ensure longer windows of clean data by removing the startle probes from the experimental design. We included the startle probes as a way to probe anxiety throughout the safe and threat blocks, however, because blink magnitude differs across conditions, the blink artifact also differs across conditions. Although beamforming can theoretically remove the artifact induced by the blink response, the only way to ensure that the blink artifact does not differentially affect the MEG signal is to remove the contaminated time periods from the analysis, or remove the startle probes from the design to begin with.

Author response image 5.Β connectivity across AAL regions during 2 second baseline prior to startle probe.**A**) Mean β connectivity across AAL regions during safe periods. **B**) Mean β connectivity across AAL regions during threat periods. **C**) Unthresholded T-test results comparing β connectivity during safe and threat conditions. Thresholded T-test results comparing β connectivity during safe and threat conditions. Labels on Y-axis correspond to regions of the AAL axis. Labels on the X-axis correspond to groups from AAL atlas (frontal, limbic, occipital, parietal, subcortical, temporal, cerebellum).**DOI:**
http://dx.doi.org/10.7554/eLife.23608.026

[Editors' note: further revisions were requested prior to acceptance, as described below.]

I thank the authors for going to such lengths on the MEG connectivity analysis. However, something has clearly gone wrong here in the analysis.

*By my assessment of the adjacency matrices presented, the case for the α and β bands look virtually identical. However this should not be the case (again see Hunt et al., 2016). It is quite hard for me to judge the spatial signature of connectivity as, unfortunately, in the pdf version the figure quality is too low resolution to read the region names on the axes. However, these matrices do not look correct to me and I would urge the authors to find out why.*

We would like to thank you for noticing this abnormality, and encouraging us to troubleshoot our MEG connectivity scripts. Indeed, we did notice an error in the analysis that affected the estimation of connectivity in the β band. When switching from α to β band we initially neglected to change the leadfield file during the construction of the virtual timeseries. This lead to an estimation of connectivity in the β band using the spatial filter optimized for the α band. This was an oversight, and has been corrected in the revised analyses. In addition, we understand that the original figures were difficult to read in the auto-generated PDF, and have included the original picture files for all adjacency matrices as a separate zip attachment.

After correcting this mistake, we re-ran the β connectivity analysis using the same method as in the previous revision. When looking at the adjacency matrices from the downsampled power envelopes, we noticed that the α and β adjacency matrices still looked very similar (See Figure 14).

Author response image 6.Adjacency matrices constructed from downsampled timeseries for the α and β bands.**DOI:**
http://dx.doi.org/10.7554/eLife.23608.027

Therefore, as recommended in your decision letter we re-ran the connectivity analyses for both the α and β bands using the non-downsampled, Hilbert-transformed, orthogonalized timeseries. (Figure 15) This analysis yielded adjacency matrices for the α and β band that looked more distinct than the analysis using the downsampled timeseries. However, the resulting correlation coefficients were much smaller from the non-downsampled timeseries.

Author response image 7.Adjacency matrices constructed from non-downsampled timeseries for the α and β bands.**DOI:**
http://dx.doi.org/10.7554/eLife.23608.028

One possibility for this reduction in the magnitude of the correlations is that some of the structure in the initial adjacency matrices resulted from variability across trials, and this variability represented a smaller proportion of the overall variability when the analysis was based on the non-downsampled timeseries. This might also explain why the adjacency matrices for the α and β bands looked so similar when constructed from the downsampled timeseries. To test this possibility, we z-scored the individual trial data before concatenating the trial timeseries and conducting the correlations. If across trial variability is sufficient to explain the larger correlation coefficients and similarity between the adjacency matrices, these effects should be eliminated by the z-scoring. This does not appear to be the case. Based on the resulting adjacency matrices, the downsampled data yield higher correlation coefficients, and the adjacency matrices from the downsampled data look similar between the α and β bands. See Figure 16.

Author response image 8.Adjacency matrices constructed from downsampled timeseries that have been converted to z-scores for the α and β bands.**DOI:**
http://dx.doi.org/10.7554/eLife.23608.029

Furthermore, the adjacency matrices from the non-downsampled data seem to lose their structure (Figure 17).

Author response image 9.Adjacency matrices constructed from non-downsampled timeseries that have been converted to z-scores for the α and β bands.**DOI:**
http://dx.doi.org/10.7554/eLife.23608.030

Together, these results suggest that the best approach is probably to use the non-downsampled, Hilbert-transformed, orthogonalized timeseries to create the adjacency matrices. However, it should be noted that this approach did not yield any significant differences in connectivity as a function of threat (See the high-resolution images included in the attached zip file). Furthermore, although the other approaches seemed to be sub-optimal, we tested whether there were any differences in connectivity as a function of threat with these as well, and were unable to identify any significant differences. In summary, we feel we have conducted a thorough analysis of the connectivity data, using methods recommended by both yourself and the reviewers, and were unable to identify any changes in connectivity as a function of threat.

Although we are happy to conduct additional analyses if requested, we feel that these analyses will likely be unfruitful, and only delay a decision on the manuscript. We understand that you are enthusiastic about the manuscript, and hope that you find it acceptable in its current form.

*In their response, the authors have suggested that they use blocks of only 2 seconds of envelope data. However they also apply downsampling prior to connectivity estimation. This to me seems silly – why apply the downsampling if you have such short blocks. It is well established that connectivity works without the downsampling so perhaps removing that step might improve the adjacency matrices. I also don't understand why such short data setments were used when in fact the bliocks are quite long. Why not just try using the whole block?*

We agree that downsampling the timeseries prior to conducting the correlation analyses was not the best approach, and have addressed this limitation (See response above).

We also agree that analyzing the entire block (rather than a series of 2 s periods) might yield more power to detect a potential difference in connectivity as a function of threat. However, we initially chose to segment the data into 2 s intervals corresponding to the pre-startle period for 2 reasons. 1) We wanted to characterize the effect of threat on pre-stimulus oscillations during the otherwise well-studied threat of shock paradigm. 2) We wanted to minimize the effect of the blink artifact on our power estimates. This is especially important in our study because startle-elicited blinks (and thus the blink artifact) are larger under threat. Although it could be argued that the adaptive beamformer spatial filter should remove noise associated with blinks from the data, we prefer a more conservative approach, and therefore chose to include only verifiably clean data in our analysis.

Again, we understand that using the data from the entire block may yield more power for the connectivity analysis, this would require re-analyzing and cleaning the data from scratch. Not only would this be time consuming, but it would also require re-running the data-cleaning ICA and the beamformer analysis, which would bring the connectivity data out of sync with the power data reported in the manuscript. If there were some indication that our current null connectivity results may be due to type 2 error, it could be argued that this re-analysis would be justified. However, given that several previous analyses yielded no effect of threat on connectivity in the MEG data, we do not feel that our results are due to type 2 error, and feel that this re-analysis of the whole-block data would not be fruitful.

Although we are happy to undertake the analysis if requested, it would require a substantial amount of time, and would significantly delay decision on the manuscript.

[Editors' note: further revisions were requested prior to acceptance, as described below.]

*I would like to thank the authors for their attempt at addressing my concerns. I'm pleased that the authors were able to spot errors in their pipeline (use of incorrect spatial filters) although it does appear that the major difference came from the down-sampling, as suggested in my previous review. Downsampling over such a small time window is obviously incorrect so these analyses should now be discarded.*

[…]

*In the light of these failed attempts, I suggest that a paragraph is added to the Discussion stating that MEG connectivity analysis would have been a useful means to probe these data; however the study design was poorly set up to make such analyses work.*

We would like to thank the reviewer for this suggestion, and we agree that this addition is necessary. In order address this concern we have added the following paragraph to the functional connectivity section of the Discussion, and edited the limitations section to be consistent with this addition.

An obvious parallel to the fMRI connectivity analysis would be to conduct a similar whole brain connectivity analysis of the MEG data. Not only would corroborating evidence from an independent imaging modality strengthen the fMRI connectivity results, but results from MEG specifically would allow for a frequency specific analysis of the effects of threat on functional connectivity (Brookes et al., 2011; Hillebrand et al., 2012). However, the current study was not optimized to reliably detect differences in MEG connectivity. In the current study we included the white noise probes as a way to obtain a quantitative measure of anxiety throughout the safe and threat blocks (Grillon et al., 1999). These white noise probes trigger the acoustic startle reflex, which varies as a function of an individual’s current level of anxiety (Grillon, 2008). Unfortunately, these reflexive blinks also inject an artifact into the MEG signal, and because the magnitude of these blinks differs across conditions, the blink artifact also differs across conditions. Although adaptive beamforming can theoretically remove the artifact induced by the blink response (Van Veen et al., 1997), the only way to ensure that the blink artifact does not differentially affect the MEG signal is to remove the contaminated time periods from the analysis, or remove the startle probes from the design at the outset. In the current study, we chose to address this limitation by extracting two-second time windows prior to each startle presentation to minimize the effect of the blink artifact on our power estimates. However, it has been shown that reliability of the MEG connectivity estimates increases as the duration of the recording increases, and durations of ~10 min or greater may be needed to maximize reliability (Liuzzi et al., 2016). Therefore, using such short intervals did not allow for the ability to obtain reliable estimates of MEG connectivity. Future studies should be conducted to address this limitation. In addition, it will be important to use appropriate correction methods to account for signal leakage between source space regions (Colclough, Brookes, Smith and Woolrich, 2015), and verify that the resulting connectivity estimates match previously published work (i.e. strong α connectivity in occipital regions and strong β connectivity linking the parietal cortex with other frontal, temporal, and occipital regions; (Hunt et al., 2016)).